# Genetic influences on circulating retinol and its relationship to human health

William R. Reay [1,2,3] ✉, Dylan J. Kiltschewskij [1,2], Maria A. Di Biase[3,4,5], Zachary F. Gerring [6], Kousik Kundu [7,8], Praveen Surendran[9,10,11], Laura A. Greco [1,2], Erin D. Clarke [12,13], Clare E. Collins[12,13], Alison M. Mondul[14], Demetrius Albanes[15] & Murray J. Cairns [1,2] ✉

Retinol is a fat-soluble vitamin that plays an essential role in many biological processes throughout the human lifespan. Here, we perform the largest genome-wide association study (GWAS) of retinol to date in up to 22,274 participants. We identify eight common variant loci associated with retinol, as well as a rare-variant signal. An integrative gene prioritisation pipeline supports novel retinol-associated genes outside of the main retinol transport complex (*RBP4:TTR*) related to lipid biology, energy homoeostasis, and endocrine signalling. Genetic proxies of circulating retinol were then used to estimate causal relationships with almost 20,000 clinical phenotypes via a phenome-wide Mendelian randomisation study (MR-pheWAS). The MR-pheWAS suggests that retinol may exert causal effects on inflammation, adiposity, ocular measures, the microbiome, and MRI-derived brain phenotypes, amongst several others. Conversely, circulating retinol may be causally influenced by factors including lipids and serum creatinine. Finally, we demonstrate how a retinol polygenic score could identify individuals more likely to fall outside of the normative range of circulating retinol for a given age. In summary, this study provides a comprehensive evaluation of the genetics of circulating retinol, as well as revealing traits which should be prioritised for further investigation with respect to retinol related therapies or nutritional intervention.

Vitamin A is an essential micronutrient that is involved in a range of important biological processes, including, vision, immune function, cell division, and neurodevelopment[1,2]. Vitamin A does not refer to a single compound, but rather to a group of compounds that encompasses retinol (all-*trans* retinol), retinoids (metabolites of retinol, such as retinaldehyde and retinoic acid), and provitamin carotenoids (beta-carotene, alpha-carotene, and beta-cryptoxanthin). Retinol is the form of vitamin A dietarily consumed from animal products, along with retinyl ester[3], while plant-based materials contain precursors termed carotenoids that can be converted to retinaldehyde[4]. Retinoic acid, an oxidised form of retinaldehyde, is a particularly potent signalling molecule that regulates the expression of thousands of genes after

binding to nuclear receptors including the retinoic-acid receptor and retinoid-X receptor subgroups[5,6].

The majority of dietary retinol is delivered to the liver, which is the primary organ responsible for its storage and metabolism. Retinol binding protein 4 (RBP4) is the major systemic transporter of retinol after hepatic secretion, facilitating delivery of retinol throughout the body[3,7,8]. RBP4 in turn complexes with the tetramer protein transthyretin (TTR), which stabilises circulating RBP4 and reduces renal filtration[9]. Notably, retinol can also be delivered directly to target tissues through other mechanisms, such as its postprandial packaging into lipid chylomicrons, as reviewed elsewhere[3].

The role of retinoid-related interventions in human disease for individuals who are not retinol deficient has been of long-standing interest. Synthetic retinoids that are structurally similar to retinol/retinoic acid are approved for dermatological indications (e.g., adapalene) and some cancers (e.g., bexarotene)[10,11], as well endogenous retinoids including all-*trans* retinoic acid (tretinoin) and 13-*cis* retinoic acid (isotretinoin) used in dermatology. There is also continued interest in repurposing both endogenous and synthetic retinoids across a range of other indications, including neuropsychiatry[2,12]. Currently, retinol supplementation is not specifically indicated unless an individual is deficient, which is rare in high-income countries, though much more common in low-income countries. Numerous observational or randomised controlled studies have explored the effects of supplementation, a high vitamin A diet, and/or measured circulating retinol in a variety of disease contexts. However, the data from these efforts have often either been null or conflicting between studies[12–16]. Despite this, recent observational evidence suggests a relationship between a greater circulating retinol abundance and lower mortality in a large, prospective 30-year follow-up study[17].

Genetics provides a powerful tool to better characterise the factors that influence the abundance of circulating retinol in serum. Moreover, genetic variants associated with retinol can be utilised to understand potential causal relationships with human health and disease, which may be informative for supplementation, dietary intervention, or repurposing retinoid pharmacotherapies[18,19]. Family studies have suggested that circulating retinol is significantly heritable[20], albeit estimated in small sample sizes. Similarly, dedicated genome-wide association studies (GWAS) of circulating retinol have also been limited to very modest sample sizes[21,22]. In 2011, Mondul et al. published findings of two genome-wide significant loci associated with retinol (N = 5006). These loci were plausibly mapped to the genes *RBP4*

and *TTR*, respectively, which form the primary serum retinol transport complex[22]. There has been comparatively little progress in further characterising the genetic architecture of retinol since that time relative to other micronutrients like vitamin D, for which large sample size GWAS (N > 400,000) have been released and extensive post-GWAS analyses performed[23,24]. The recent adoption of untargeted high-throughput metabolomics platforms with coverage for retinol in some existing genotyped cohorts presents a new opportunity to boost statistical power.

In this work, we perform the largest GWAS of circulating retinol to-date (Fig. 1). We identify novel loci that are associated with retinol beyond the RBP4:TTR transport complex, including signals that plausibly are involved in pathways such as lipid metabolism. These data are leveraged to investigate the effect of retinol on thousands of human health phenotypes, revealing relationships of potential clinical significance.

## Results

### The common and rare variant genetic architecture of circulating retinol

We integrated common and rare variant data from up to 22,274 individuals of European ancestry in our discovery meta-analyses to estimate genetic effects on circulating retinol (Fig. 2A, B, Online Methods). Firstly, variants from the INTERVAL and METSIM studies were meta-analysed (N_Meta = 17,268 – termed METSIM + INTERVAL, Fig. 1, Online Methods). After harmonisation, there were 8,173,975 common and 5,091,050 rare [minor allele frequency (MAF) < 1%] overlapping variants between INTERVAL and METSIM, respectively. As retinol effect sizes were estimated in the same units in both studies [plasma standard deviation (SD) units, quantified by the same instrument], we conducted both a fixed-effects inverse-variance weighted (IVW)

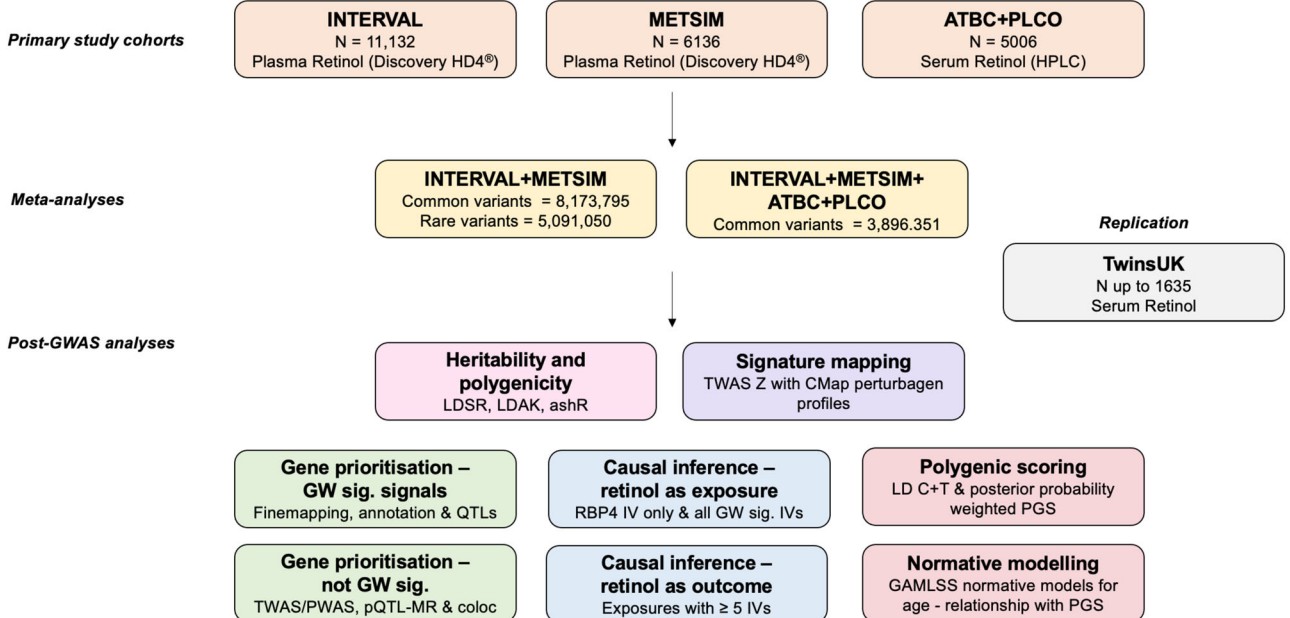

**Fig. 1 | Overview of study design for the genome-wide meta-analysis of circulating retinol.** In this study, we used three input datasets – INTERVAL, METSIM, and ATBC/PLCO. Our primary meta-analysis was INTERVAL meta-analysed with METSIM. These two studies measured plasma retinol using the same platform and both had excellent genome-wide coverage that allowed the analysis of millions of overlapping common and rare variants. ATBC + PLCO was imputed using a much older imputation panel (HapMap2) and even after summary statistics imputation there were markedly fewer variants available. As a result, we performed a separate meta-analysis that included ATBC + PLCO, which had a larger sample size but fewer variants available genome-wide. This dual strategy was adopted to balance variant

coverage (METSIM + INTERVAL), which improves discovery power, with the power afforded by increased sample size (METSIM + INTERVAL + ATBC + PLCO). Replication for the lead SNPs was then attempted in up 1635 participants in TwinsUK, with replication performed at each of the three-retinol measurement timepoints, respectively, for the twin pairs separated. Several post-GWAS analyses were then performed. These included: estimates of heritability and polygenicity, gene prioritisation, causal inference, signature mapping, polygenic score development for circulating retinol, as well as the integration of this retinol polygenic score with normative modelling derived deviations.

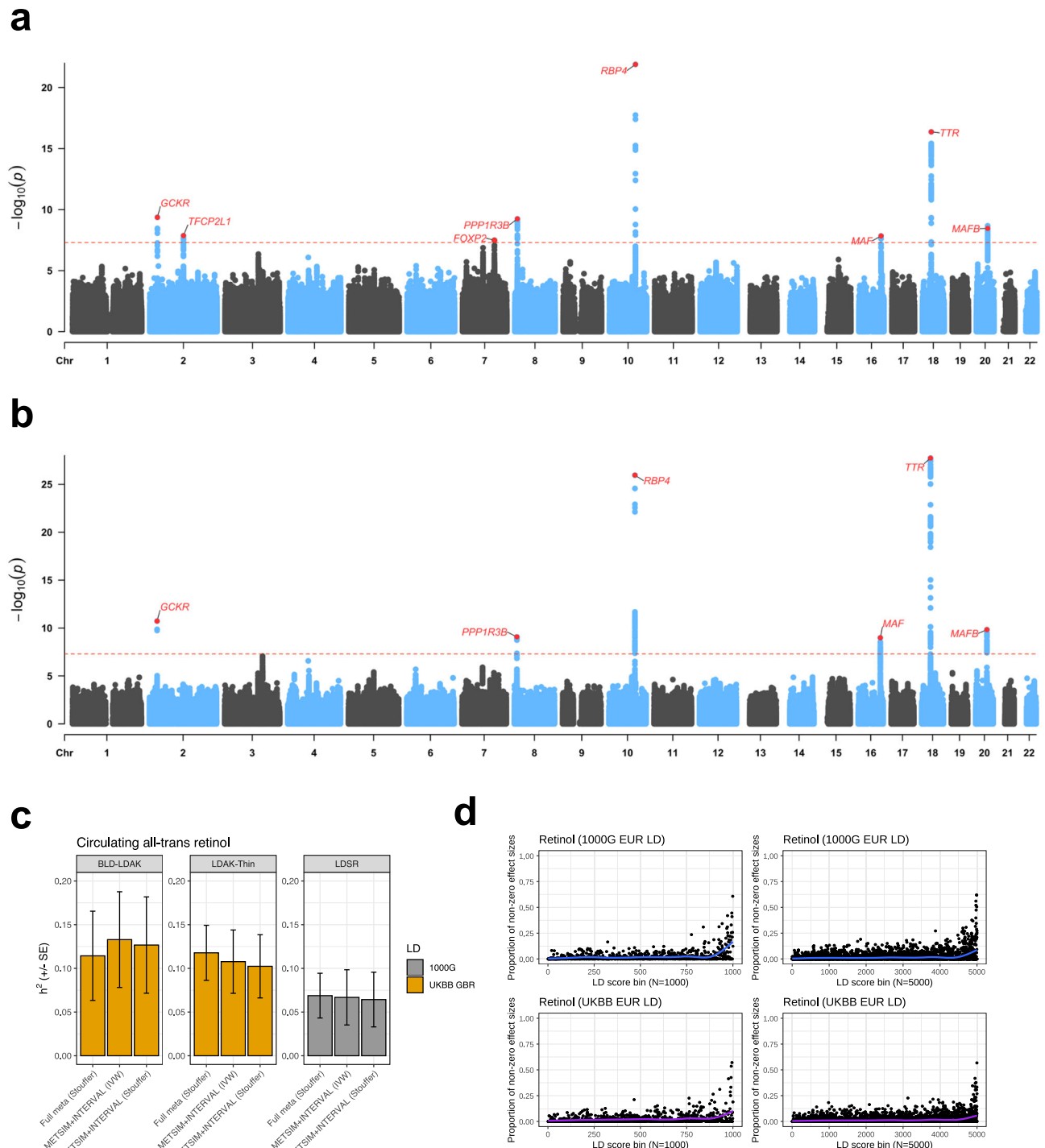

**Fig. 2 | Common variant influences on circulating retinol. a** Manhattan plot of the meta-analysis of common variants shared between the INTERVAL and METSIM cohorts (Stouffer's sample size weighted meta-analysis). Variant-wise -log$_{10}$ P-values for association are plotted, with the dotted red line denoting genome-wide significance. The closest genic transcription start site is labelled for each lead SNP. Constituent GWAS performed using multiple-linear regression. **b** Manhattan plot, as above, for the larger sample size meta-analysis that includes the ATBC and PLCO cohorts, but with fewer variants available for meta-analysis. Constituent GWAS performed using multiple-linear regression. **c** Estimates of SNP heritability of retinol (h$^2$), with the error bars denoting the standard errors of the estimates. The first two panels denote estimates using the BLD-LDAK model and the LDAK-thin model, respectively, both using LD tagging files from the Great British ancestry

participants in the UK Biobank. The last panel estimates heritability using the LDSR model with LD from the 1000 genomes European participants. Estimates were for the METSIM + INTERVAL meta-analyses (Stouffer's and IVW), as well as the larger meta-analysis including ATBC/PLCO. **d** Empirical Bayes' estimation of non-null effects on retinol genome-wide, stratified by bins of ascendingly sorted LD score by magnitude. The LD score bins were different for each panel – 1000 bins, 1000 genomes European LD scores (top left); 5000 bins, 1000 genomes European LD scores (top right); 1000 bins, UKBB white British LD scores (bottom left); 5000 bins, white British LD scores (bottom right). Each point represents the proportion of non-null effect sizes for that bin, with the trendline estimated using a generalised additive model for the relationship between ascending LD score bin and the proportion of non-null effects.

**Table 1 | Genome-wide significant common loci associated with retinol (METSIM + INTERVAL)**

| Lead SNP (Stouffer) | Locus | Closest Gene (TSS) | EA/NEA | EAF NFE/FIN | Z | $P_{GWAS}$ | $P_{Het}$ | Reported by Mondul et al.[22] |
|---|---|---|---|---|---|---|---|---|
| rs1260326 | 2:27598097-27752871 | GCKR | T/C | 0.409/0.358 | 6.242 | $4.32e^{-10}$ | 0.023 | No |
| rs34898035 | 2:122078406-122084285 | TFCP2L1 | A/G | 0.042/0.027 | −5.677 | $1.37e^{-8}$ | 0.190 | No |
| rs11762406 | 7:114014488-114286611 | FOXP2 | A/C | 0.092/0.085 | −5.526 | $3.28e^{-8}$ | 0.104 | No |
| rs6601299 | 8:9167797-9224907 | PPP1R3B | T/C | 0.17/0.10 | −6.197 | $5.76e^{-10}$ | 0.004 | No |
| **rs10882283** | **10:95295876-95360964** | **RBP4** | **A/C** | **0.622/0.664** | **9.789** | **$1.26e^{-22}$** | **0.192** | **Yes** |
| rs12149203 | 16:79696939-79756197 | MAF | C/G | 0.708/0.726 | 5.668 | $1.45e^{-8}$ | 0.841 | No |
| **rs1667226** | **18:29134171-29190174** | **TTR** | **A/T** | **0.481/0.507** | **−8.405** | **$4.27e^{-17}$** | **0.06** | **Yes** |
| rs6029188 | 20:39142516-39234223 | MAFB | A/G | 0.637/0.662 | −5.908 | $3.47e^{-9}$ | 0.147 | No |

Lead SNP based on statistical significance from sample size weighted (Stouffer) meta-analysis. Loci boundaries as defined by FUMA (hg19 coordinates). Constituent GWAS was performed using multiple linear regression. The closest gene by transcription start site (TSS) is listed. EA = effect allele, that is, allele to which the effect size relates, NEA = non-effect allele. The effect allele frequency (EAF) is given from gnomAD v2.1.1 for non-Finnish Europeans (NFE) and Finnish Europeans (FIN). Z-score from the Stouffer meta-analysis. The $P_{GWAS}$ (statistical significance of association) and $P_{Het}$ (Heterogeneity between input effect sizes P value derived from Cochran's Q) are also from the Stouffer meta-analysis. Bolded loci are known retinol signals. The far right column denotes whether they were genome-wide significant in the previous Mondul et al. retinol GWAS (ATBC + PLCO)[22].

meta-analysis and a sample-size weighted meta-analysis of Z scores (Stouffer's method). Secondly, we conducted an additional meta-analysis ($N_{Meta}$ = 22,274) which also included data from two other studies (ATBC and PLCO), termed the METSIM + INTERVAL + ATBC + PLCO meta-analysis. However, there were markedly fewer variants available in this meta-analysis after imputing the ATBC and PLCO summary statistics and harmonising with METSIM + INTERVAL ($N_{Var}$ = 3,896,351, Online Methods). As a result, we focused on the METSIM + INTERVAL meta-analysis as the primary discovery dataset.

Considering common variants from the HapMap3 panel (MAF > 0.05, outside of the major histocompatibility complex (MHC) region), we observed a relatively subtle inflation of retinol signals across the genome as indexed by the mean $\chi^2$ statistic, with mean $\chi^2$ values around 1.03-1.04, regardless of the meta-analysis considered (Supplementary Data 1). Common variant SNP heritability ($h^2_{SNP}$) was firstly estimated using the linkage disequilibrium (LD) score regression (LDSR) approach and the 1000 genomes European reference panel. SNP heritability point estimates from this approach were between 6%-7% and nominally statistically significant ($P < 0.05$), although with somewhat large standard errors (2.6%-3.2%) (Fig. 2C, Supplementary Data 2). SNP heritability estimates of circulating retinol increased to between 10%-13%, but were still noisy, using two alternate models to estimate $h^2_{SNP}$ and the UK Biobank (UKBB) as the LD reference (Fig. 2C, Online Methods, Supplementary Data 2). Partitioned $h^2_{SNP}$ across tissues and cell-types demonstrated nominal enrichment in logical contexts given the known biology of retinol homoeostasis such as liver, adipose, pancreas, and blood (Supplementary Fig. 1). The somewhat large $h^2_{SNP}$ standard errors are likely a product of sample size; however, we then conducted further analysis to explore the extent of the polygenic signal associated with retinol across the genome using an Empirical Bayes' method (Online Methods). This method was utilised to model the number of non-null effects on retinol genome-wide stratified by bins of LD scores that index the extent of LD any given variant exhibits with other variants (Fig. 2D). Across all LD score bins, we estimated that the mean fraction of common variants across the genome with non-null effects on retinol was between 1.4% to 2.4%, depending on the modelling parameters used. In line with expectation, the proportion of non-null retinol effects was very high (> 50%) when considering the variants that display the most extensive LD (highest LD score bins). The application of these analyses to two GWAS of another vitamin (25-hydroxyvitamin $D_3$) with either comparable or much larger sample sizes[23,25], suggested that the less-polygenic architecture of retinol observed in this study may become more diffuse across the genome with greater sample sizes, in line with many other quantitative traits (Supplementary Fig. 2). However, we caution that further investigation will be required as these data become available to confirm this.

Next, we processed the common variant results of both meta-analyses to identify genome-wide significant loci associated with circulating retinol ($P_{GWAS} < 5 × 10^{-8}$). In the primary discovery meta-analysis (METSIM + INTERVAL, Stouffer's method), we uncovered eight genome-wide significant loci, six of which were not reported in the previous Mondul et al. retinol GWAS (Table 1). The absolute value of the effect sizes ($\beta$) of these lead SNPs were between 0.066 and 0.172 SD in circulating retinol per effect allele, as derived from the IVW meta-analysis. We observed minimal heterogeneity between the two cohorts for these lead SNPs, although heterogeneity was slightly more marked for rs6601299 (Supplementary Fig. 3,4). Replication was then attempted in the TwinsUK cohort (N up to 1635, Online methods). Considering the mean association across all timepoints retinol was measured, as well as the twin pairs separately, 7 out of the 8 lead SNPs in the loci had effect sizes in the same direction, which was greater than expected by chance alone (Binomial $P = 0.035$, Supplementary Data 3). Of these, the TTR, GCKR, FOXP2, and PPP1R3B lead SNPs, denoted as such based on their closest TSS, were at least nominally significant at one or more of the measured timepoints.

We also investigated the effect of retinol-associated lead SNPs on factors associated with dietary intake of retinoids. Using a GWAS of retinol intake derived from a self-reported 24-hour dietary recall in the UK Biobank (UKBB, N = 62,991)[26], we found no evidence to suggest that any of the effect of these genetic signals on circulating retinol is mediated through influencing dietary intake behaviours (Supplementary Data 4, Supplementary Fig. 5).

In the larger sample-size meta-analysis with fewer variants available across all input datasets (METSIM + INTERVAL + ATBC + PLCO), six of the eight genome-wide significant loci from the smaller meta-analysis were available to test for association. Five of the six loci available became more statistically significant in this larger meta-analysis, whilst the chromosome 8 locus obtained a very similar level of statistical significance in both meta-analyses (Table 2). It should be noted that there was relatively large heterogeneity for the lead SNP at the TTR locus on chromosome 18 locus between cohorts (Supplementary Fig. 6). This was due to a more significant association in the ATBC + PLCO GWAS, although this region was still very strongly associated in both METSIM and INTERVAL in the same direction (Supplementary Fig. 6).

We also estimated rare variant (frequency <1%) effects on circulating retinol using variants available in the METSIM + INTERVAL meta-analysis. Despite relatively low power to detect rare variant association, we identified a genome-wide significant rare variant signal on chromosome five (chr5:86765041:T:C, dbSNP ID: rs138675130) associated with reduced circulating plasma retinol per C allele (−0.441 SD, $SE = 0.0709$, $P = 6.37 × 10^{-9}$). There was no significant heterogeneity

between the contributing studies for this signal. The frequency of this C allele in Europeans (gnomAD v.3.1.2) ranges from 0.5% in non-Finnish Europeans to 0.8% in Finns, and whilst it is marginally rarer in Africans and South Asians, it is entirely absent in the East Asian and Middle Eastern populations in the gnomAD database. The variant is intergenic and in FinnGen release 8, the C allele was associated at phenome-wide significance ($P < 1 \times 10^{-5}$) with increased odds of benign neoplasm of the eye and adnexa, as well as whooping cough. The closest canonical transcription start site to this variant is that of *COX7C*, which encodes a subunit of a terminal component of the mitochondrial respiratory chain. We also uncovered several suggestively significant rare variant signals ($P < 1 \times 10^{-5}$), including three non-synonymous variants in the genes *FREM2*, *NAXD*, and *CHD1* (Supplementary Data 5). Of these, the rare *NAXD* non-synonymous allele suggestively associated with lower retinol (rs3742192) had some in silico evidence to suggest deleteriousness (Supplementary Data 5), although it is classed as benign in ClinVar. Finally, to boost power, we also statistically aggregated rare variants to genes (Online Methods). There were no significant retinol genes after Bonferroni correction of these gene-level association results ($P < 3.02 \times 10^{-7}$); however, there were two novel suggestively associated genes ($P < 3.02 \times 10^{-5}$), *GALM* and *ZDHHC18* (Supplementary Data 6).

## Prioritisation of retinol-associated genes reveals mechanisms of genetic influence on circulating retinol

In common variant loci, prioritising causal genes can be difficult due to confounding factors like linkage disequilibrium. For circulating retinol, we employed a multi-faceted approach to prioritise genes that are confidently associated. Firstly, we sought to interrogate the eight genome-wide significant loci uncovered in the main discovery meta-analysis (METSIM + INTERVAL) to uncover plausible causal genes. This was achieved by adapting an integrative pipeline developed in previous work that considers annotation, probabilistic finemapping, integrative scoring, and in silico prediction (Online Methods, Supplementary Data 7). We describe these results further for each locus in supplementary note 1 but summarise the prioritisation evidence forthwith. There was quite consistent evidence in four loci for a likely causal gene (*GCKR*, *FOXP2*, *RBP4*, and *TTR*). *RBP4* (chromosome 10 locus) and *TTR* (chromosome 18 locus) were previously associated with retinol in the only other dedicated GWAS of this trait and form the complex that transports retinol in serum[22], thereby, having a direct biological link to retinol abundance. *GCKR*, a gene encoding a protein that binds to and regulates the key metabolic enzyme glucokinase, was confidently the causal gene for the locus on chromosome 2 with the rs1260326 lead SNP. This gene is known to have a large and varied metabolic association profile due to the role of glucokinase in glycaemic and lipid-related processes, amongst others[27–29]. Interestingly, the lead SNP (rs1260326), and most likely causal variant derived from probabilistic finemapping, was a common missense allele, whereby the retinol increasing *T* allele corresponds to a substitution of leucine for proline in the GCKR protein. Previous experimental work suggests that this variant impacts GCKR affinity for glucokinase[27,30].

The transcription factor gene *FOXP2* was also strongly supported by multiple lines of evidence as a causal gene for the locus on chromosome seven. The role of this gene in the brain and in relation to neurological phenotypes like language has been extensively studied[31], but less so in the periphery despite relatively high expression across many different systemic tissues. Therefore, we analysed RNA-sequencing data of *FOXP2* overexpression in a cell-line not derived from the central nervous system (human osteosarcoma epithelial cell line) and revealed the transcriptional correlates of *FOXP2* overexpression were enriched for a broad range of pathways related to factors including extracellular matrix biology, glycosylation, and interleukin signalling, amongst many others (Supplementary Data 8-11, Online Methods)[32].

**Table 2 | Genome-wide significant common loci associated with retinol in the larger meta-analysis (METSIM + INTERVAL + ATBC + PLCO)**

| Lead SNP (Stouffer) | Locus | Closest gene (TSS) | EA/NEA | Z-score (METSIM + INTERVAL) | Z-score (METSIM + INTERVAL + PLCO + ATBC) | $P_{GWAS}$ | $P_{Het}$ | Reported by Mondul et al.[22] |
|---|---|---|---|---|---|---|---|---|
| rs1260326 | 2:27598097-2775871 | GCKR | T/C | 6.242 | 6.714 | 1.90e-11 | 0.06 | Yes |
| rs6601299 | 8:9167797-9224907 | PPP1R3B | T/C | -6.197 | -6.137 | 8.41e-10 | 0.004 | Yes |
| rs11187547 | 10:95279771-95360964 | RBP4 | A/G | 8.769 | 10.691 | 1.12e-26 | 0.11 | No |
| rs11865979 | 16:79696939-79756197 | MAF | T/C | 5.575 | 6.103 | 1.04e-9 | 0.887 | Yes |
| rs4799581 | 18:29068068-29230411 | TTR | T/C | -7.998 | -11.065 | 1.85e-28 | 9.72e-6 | No |
| rs6029188 | 20:39152458-39234223 | MAFB | A/G | -5.908 | -6.407 | 1.48e-10 | 0.299 | Yes |

Lead SNP based on statistical significance from sample size weighted (Stouffer) meta-analysis for SNPs available in this extended analysis. Constituent GWAS was performed using multiple linear regression Loci boundaries as defined by FUMA (hg19 coordinates). The closest gene by transcription start site (TSS) is listed. EA = effect allele, that is, allele to which the effect size relates, NEA = non-effect allele. The Z score is given for the full extended meta-analysis as well as the smaller METSIM + INTERVAL analysis in terms of sample size. The $P_{GWAS}$ (statistical significance of association) and $P_{Het}$ (Heterogeneity between input effect sizes P value derived from Cochran's Q) are also from the Stouffer meta-analysis from the extended meta-analysis. Bolded loci are known retinol signals. The last column refers to whether they were genome-wide significant in the previous Mondul et al. retinol GWAS (ATBC + PLCO)[22].

The remaining four loci exhibited less clear evidence of which gene to prioritise, although all point to potentially interesting functional mechanisms. On chromosome eight, there is some evidence to support *PPP1R3B* as a gene that influences retinol, which encodes a catalytic subunit of the phosphatase PP1 that is implicated in relevant metabolic processes like glycogen synthesis[33]. However, other lines of evidence point to the role of long-noncoding RNA in this locus. The loci on chromosomes 16 and 20 are noteworthy as the closest transcription start sites to the respective lead SNPs are two transcription factors (TF) from the *Maf* family (*MAF* and *MAFB*). Interestingly, *MAFB* has been shown to regulate both *TTR* and *RBP4* expression in various tissue contexts from human or murine studies[34,35]. As only some lines of evidence support these two TF, further functional characterisation of these two loci is warranted. Interestingly, in the locus on chromosome 16, two of the other genes with some evidence for a retinol-related function (*MAFTRR* and *LINC01229*) have been shown experimentally to regulate *MAF* expression and are also associated with other biochemical traits like urate, further supporting the role of the *Maf* family on retinol abundance in serum[36]. Finally, the remaining locus on chromosome 2 (2:122078406-122084285) had the least interpretable functional prioritisation results. The closest transcription start site to the lead SNP was another TF (*TFCP2L1*) that has broad physiological roles, including in the kidney[37].

We then sought to expand our scope for gene discovery beyond genome-wide significant retinol-associated loci through further integration of genetics with transcriptomics and proteomics (Online Methods, Supplementary Data 12-13). Firstly, we leveraged multivariate models of genetically regulated expression (GReX) to perform a transcriptome-wide (liver, whole blood, adipose, small intestine, pancreas, and breast mammary tissue) and proteome-wide (plasma) association study (TWAS/PWAS) of circulating retinol. Tissues for the TWAS were selected based on prior knowledge of retinol biology and the results of the partitioned heritability analyses (Online Methods), whilst plasma was the only tissue available for PWAS. After applying multiple-testing correction to the TWAS and PWAS individually (FDR < 0.05) and testing whether there was a shared causal variant via colocalisation [Posterior probability (PP) of a shared causal variant (H4), $PP_{H4} > 0.8$], we identified strong evidence of four additional retinol-associated genes outside of genome-wide significant loci (at least +/− 1 megabase away). These were as follows: *MLXIPL*, which binds to carbohydrate response elements to regulate triglycerides[38–40]; *GSK3B*, a gene that encodes a member of the glycogen synthase kinase family involved in metabolism and glycaemic homoeostasis[41]; the tankyrase gene (*TNKS*) implicated in processes like Wnt signalling[42]; and *INHBC*, part of the inhibin family of proteins with important endocrine functionality[43]. Genetically predicted mRNA expression of *MLXIPL* in adipose, pancreas, and breast mammary tissue was inversely associated with circulating retinol levels. Conversely, TWAS analyses revealed that genetically predicted expression of *GSK3B* and *TNKS* was positively associated with circulating retinol levels. Finally, genetically predicted plasma protein expression of INHBC showed a positive correlation with retinol levels ($Z_{PWAS} = 4.72$). We then used a more conservative approach whereby finemapped variants that putatively influence protein expression (pQTLs) were used as instrumental variables (IV) to estimate the causal effect of plasma proteins on retinol using Mendelian randomisation (MR, Online Methods, Supplementary Data 14). We applied the same filters to the results (FDR < 0.05 and $PP_{H4} > 0.8$). The colocalisation tests assist to detect proteomic effects on retinol that have higher confidence; however, as we are mostly using a single finemapped IV per protein, we do not have access to the same suite of sensitivity analyses that are usually deployed in a polygenic MR approach which uses multiple IVs. These pQTL-MR analyses further supported that upregulated INHBC likely increases serum retinol, with each SD increase in plasma protein expression associated with a small but highly statistically significant impact on circulating retinol [0.05 SD increase in retinol per SD in INHBC expression, 95% CI: 0.03, 0.07]. In line with expectation, pQTL-MR, and subsequent colocalisation, further genetically validates that elevated RBP4 protein abundance correspondingly increases serum retinol with somewhat large effect (0.6 [95% CI: 0.48, 0.72] SD in retinol per SD in plasma RBP4 expression). There was also evidence that RBP4 protein expression and retinol colocalise under the hypothesis of a single causal variant ($PP_{H4} = 1$). Considering the eight genes prioritised in this and the previous section (*RBP4, GCKR, FOXP2, TTR, MLXIPL, GSK3B, TNKS, INHBC*), we found that these genes exhibited upregulated expression in the liver ($P_{Adjusted} < 0.05$) upon analysing data from 54 GTEx tissues. This further consolidates the salience of hepatic processing to genetic influences on circulating retinol abundance. Pathway analyses of these genes demonstrated that they were enriched amongst factors such as those involved in carbohydrate metabolism (Supplementary Data 15).

## Evidence of causal effects of retinol across the human clinical phenome using retinol binding protein 4 variation as a genetic instrument

The role of retinol in human health and disease has been of long-standing interest. However, most evidence has been observational, limiting the ability for causal inference. Moreover, randomised controlled trials (RCT) of interventions like retinol supplementation and endogenous/synthetic retinoids have only been performed for a small fraction of the traits implicated through observational studies. We sought to increase our understanding of causal effects of genetically predicted retinol on human health by leveraging genetic variants associated with retinol uncovered in this study as IVs. Given certain assumptions are met, these genetic proxies of retinol can be utilised to estimate causal effects of circulating retinol at scale using Mendelian randomisation (MR) (Online Methods). Firstly, we utilised a single IV in *RBP4* (rs10882283), as this gene has a clear and unambiguous association with circulating retinol levels, and therefore, is less likely to be prone to horizontal pleiotropy than other retinol-associated loci. We do caution, however, that *RBP4* does exert some other functionality that may not be directly related to retinol transport[44], although this gene is still likely the best available single IV associated with circulating retinol at genome-wide significance. We utilised this *RBP4* IV (MET-SIM + INTERVAL meta-analysis effect size in SD units) to estimate the effect of retinol on over 19,500 outcomes in the IEUGWAS database (IEUGWASdb, Online Methods, Supplementary Data 16). IEUGWASdb collates and harmonises GWAS summary data from GWAS catalog, consortium based GWAS, UK Biobank GWAS, and several other sources. After multiple-testing correction (FDR < 0.01), genetically predicted retinol was found to putatively exert causal effects on outcomes including several lipid traits, leucocyte counts (total leucocytes and neutrophils), reticulocytes, optic disc area, and resting-state connectivity of a functional MRI (fMRI) derived network edge. For instance, each SD in circulating retinol was associated with a −0.1 SD [95% CI: −0.14, −0.06] decrease in leucocyte count, whilst this unit retinol increase was estimated to increase optic disc area by 0.22 SD [95% CI: 0.12, 0.32]. We further interrogated a representative subset of these associations using colocalisation to test if the signals are driven by a shared causal variant in *RBP4* (Fig. 3, Supplementary Data 17). Colocalisation strongly supported that the effect of circulating retinol on leucocyte count, optic disc area and the edge in the fMRI network arises from a shared causal variant in *RBP4*. In contrast, the effect of retinol on lipids through *RBP4* was shown to likely arise from linkage due to the proximal gene *FFAR4* that encodes a free-fatty acid receptor. Therefore, the lipid findings may not represent a causal impact of circulating retinol, at least through *RBP4*, but rather the influence of *FFAR4*, although further investigation is warranted, particularly using analyses that consider multiple causal variants at this locus.

A limitation of the above phenome-wide analyses is that the multiple-testing burden that arises from the inclusion of over 17,000

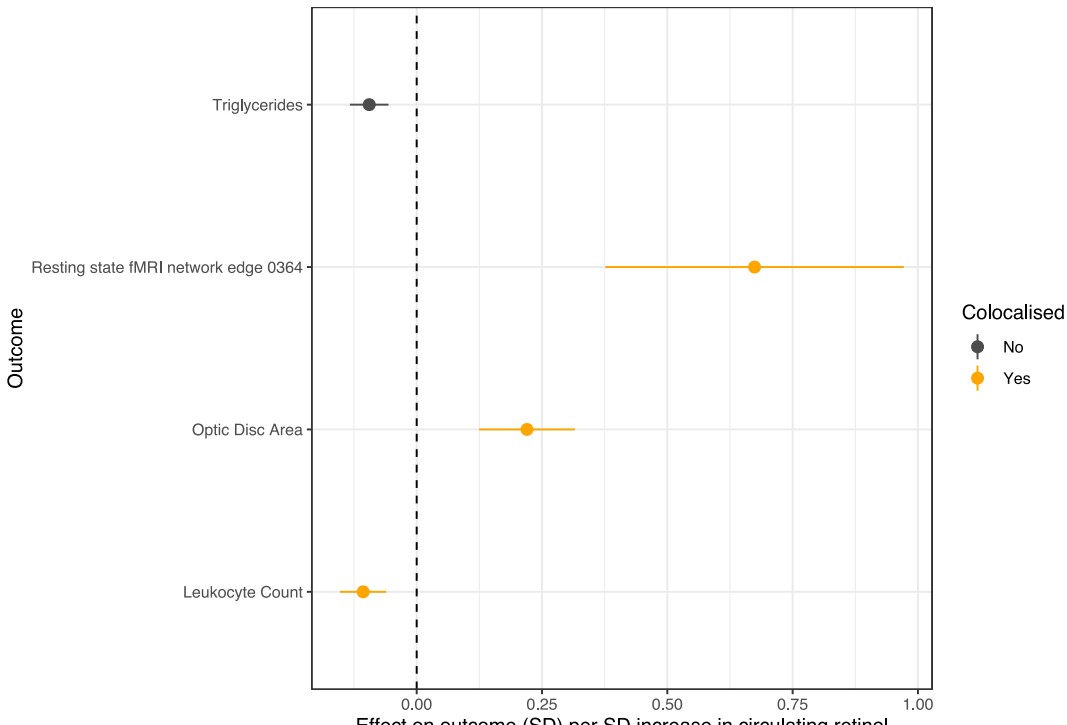

**Fig. 3 | Representative subset of significant causal estimates of circulating retinol using the RBP4 instrumental variable (IV) subjected to colocalisation.** The effect size units are the beta per standard deviation (SD) increase in circulating retinol. The error bars denote 95% confidence intervals. Traits highlighted orange demonstrate strong evidence of a single shared causal variant with circulating retinol in the *RBP4* region ($PP_{H4} > 0.8$).

traits may obscure retinol effects on binary disease phenotypes, as these are usually less powered than GWAS of continuous traits. To overcome this, we also estimated the effect of retinol using the *RBP4* IV on 1141 binary endpoints with at least 1000 cases in FinnGen release 8 (not featured in IEUGWASdb), allowing a phenome-wide analysis of electronic health record derived binary outcomes. However, there were no traits that survived multiple-testing correction. The most statistically significant result was some nominal evidence of a protective effect of circulating retinol on the odds of carpal tunnel syndrome – OR = 0.682 [95% CI: 0.558, 0.860], $P = 2.2 \times 10^{-4}$, q = 0.25.

## Exploratory analysis of phenome-wide effects of retinol using multiple instrumental variables

To boost power for discovery of causal retinol effects, we then utilised non-palindromic, independent (LD $r^2 < 0.001$) genome-wide significant SNPs as IVs (Online Methods). We do caution that due to the immense pleiotropy of some IVs, such as the SNP in *GCKR*, these results must be interpreted carefully. As a result, we aimed to implement a pipeline that refines the trait pairings with the strongest evidence. Firstly, we used the inverse-variance weighted estimator with multiplicative random effects (IVW-MRE) to estimate the effect of retinol on IEUGWASdb outcome GWAS for which at least six of the IVs were available (> 17,000 outcome phenotypes). There was moderate positive correlation between the IVW-MRE and single *RBP4* estimates across all traits tested (r = 0.43, Supplementary Fig. 7). Whilst well powered, the IVW-MRE assumes all IVs are valid, which is unlikely in practice. Therefore, we developed a pipeline to prioritise the most confident causal relationships that survive multiple-testing correction considering the IVW-MRE estimates (FDR < 0.01, Online Methods, Fig. 4A, Supplementary Data 18–20). This was comprised of three tiers, with Tier #1 being the highest level of evidence. Retinol/outcome pairings that were assigned a tier had to exhibit no significant heterogeneity between IV-outcome effects, a non-significant MR-Egger intercept (which screens for unbalanced pleiotropy), and not be driven by a single IV. Four

additional MR methods with differing assumptions were then applied in this study (Online Methods). From the trait pairings that passed the above heterogeneity and pleiotropy filters, Tier #1 traits were those for which all five methods were nominally statistically significant (P < 0.05), whilst Tier #2 traits had 4/5 significant methods, and Tier #3 traits 3/5 methods significant. There were no Tier #1 retinol/outcome trait pairings, but several Tier #2 and Tier #3 trait pairings (Fig. 4B, C), with all of them directionally consistent with the estimates from the single *RBP4* IV, supporting their validity. Before formally reporting trait pairings as either Tier #2 or Tier #3 in the manuscript, we excluded traits that would have been assigned a tier with non-European ancestry (Age hay fever, rhinitis or eczema diagnosed, cereal type, and sitting height ratio), and ensured that the GWAS reported were large and well powered relative to the outcome trait. Broadly, we found evidence that genetically predicted retinol may increase body fat-related measures, resting-state fMRI connectivity of several network edges, as well as food consumption phenotypes related to carbohydrates. Retinol also exhibited evidence of a relationship with the cortical thickness and surface area of several brain regions, as well as microbiome composition and keratometry measurements. These results were dominated by continuous traits, for which we are better powered, however, there were some binary traits assigned Tier #2 or Tier #3 evidence. Specifically, genetically proxied retinol was associated with decreased odds of coxarthrosis (arthrosis of the hip), whilst it was associated with adverse asthma/COPD medication effects and dental problems. We caution that all these inferred associations require further investigation, and should be treated with requisite caution, as reviewed elsewhere[19,45,46]. GWAS of these outcome traits with greater sample size will also no doubt emerge in the future, and these findings should be assessed for consistency using those larger studies. Moreover, the power and assumptions of each MR test is also a key consideration when further interrogating these reported trait pairings – for instance, the MR-Egger approach is relatively low-powered compared to other methods used here but is unbiased in the presence of pleiotropy under

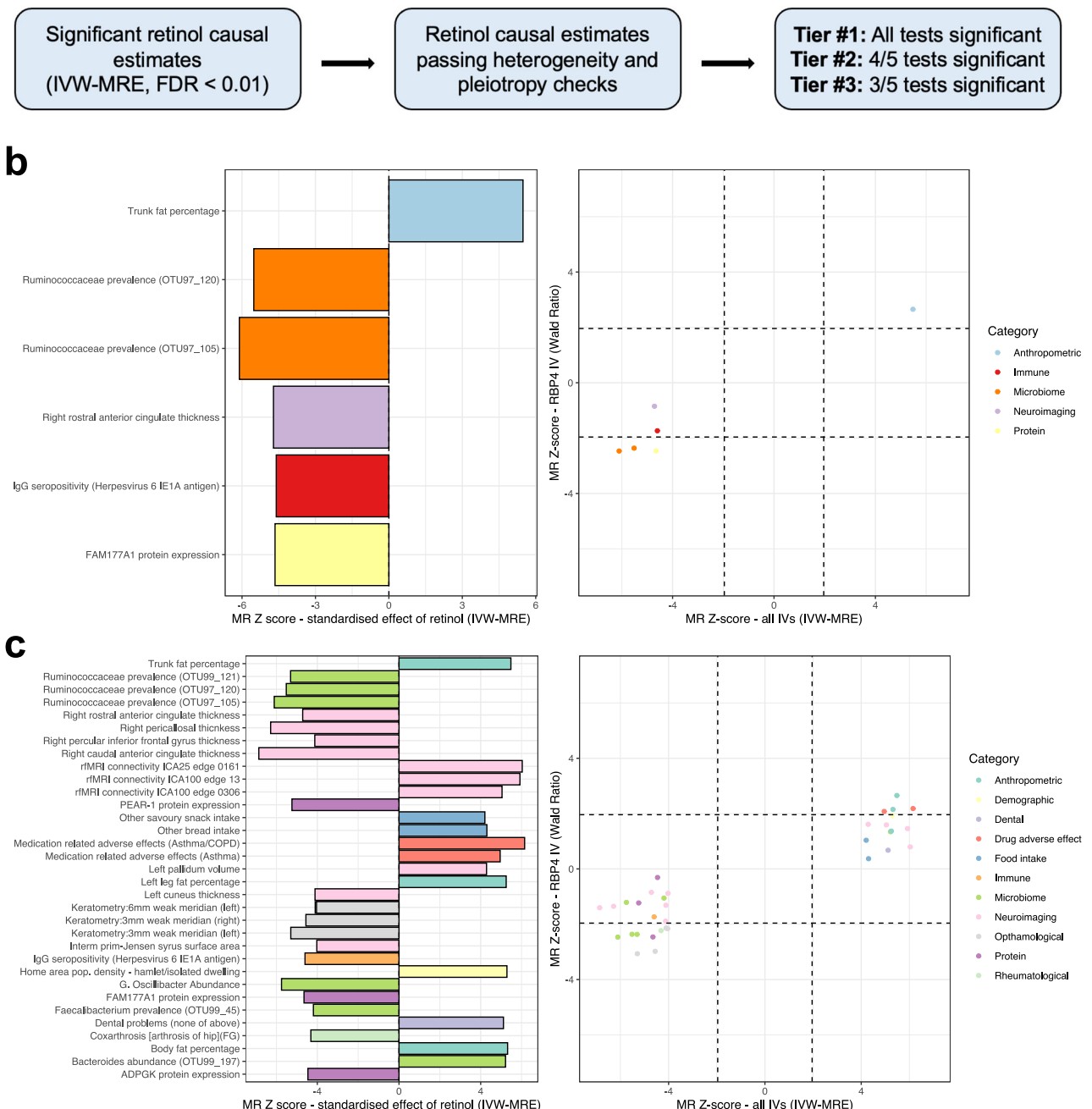

**Fig. 4 | Estimated causal effects of genetically predicted circulating retinol across the human clinical phenome. a** Prioritisation pipeline overview for retinol causal estimates [inverse-variance weighted estimator with multiplicative random effects (IVW-MRE)] that survive multiple-testing correction (FDR < 0.01). These estimates are then subjected to tests for heterogeneity and pleiotropy (Online Methods), with a tier then assigned based on how many of the five Mendelian randomisation (MR) methods applied are at least nominally statistically significant. In panels **b** and **c**, the left-hand plot denotes the Z-score (beta/SE) from the MR IVW-MRE estimates. Positive Z scores denote a positive IVW-MRE estimate of the effect of circulating retinol on that trait, and vice vera. The traits are coloured by their broad phenotypic category. The right-hand plot visualises the Z score using the IVW-MRE model verses that of the MR estimate using the RBP4 IV alone (Wald Ratio). The dotted lines approximately represent nominal statistical significance (P < 0.05). In panel **b**, just tier #2 traits are plotted (Online Methods), whilst panel **c** plots both tier #2 and tier #3 traits.

the InSIDE (Instrument Strength Independent of Direct Effect) assumption. Another consideration of this approach, which leverages multiplicative random effects with relatively few IVs (<10), is the potential influence of residual standard errors below 1 on the estimation of the IVW standard errors. We see some evidence of this impact on the IVW-MRE estimates for Tier #2 and Tier #3 traits – as these traits exhibit no significant heterogeneity between IV estimates. Specifically,

whilst all fixed effect IVW results are still highly statistically significant for these traits, they have larger standard errors than the IVW-MRE, indicative of residual standard errors <1. This is a function of the MRE not scaling the standard error of the IVW by the model residual standard error like in the fixed effects model. We discuss this further in Supplementary note 2 and in Supplementary Fig. 8. However, these issues only impact the *P*-value of the IVW-MRE relative to that of the

**a**

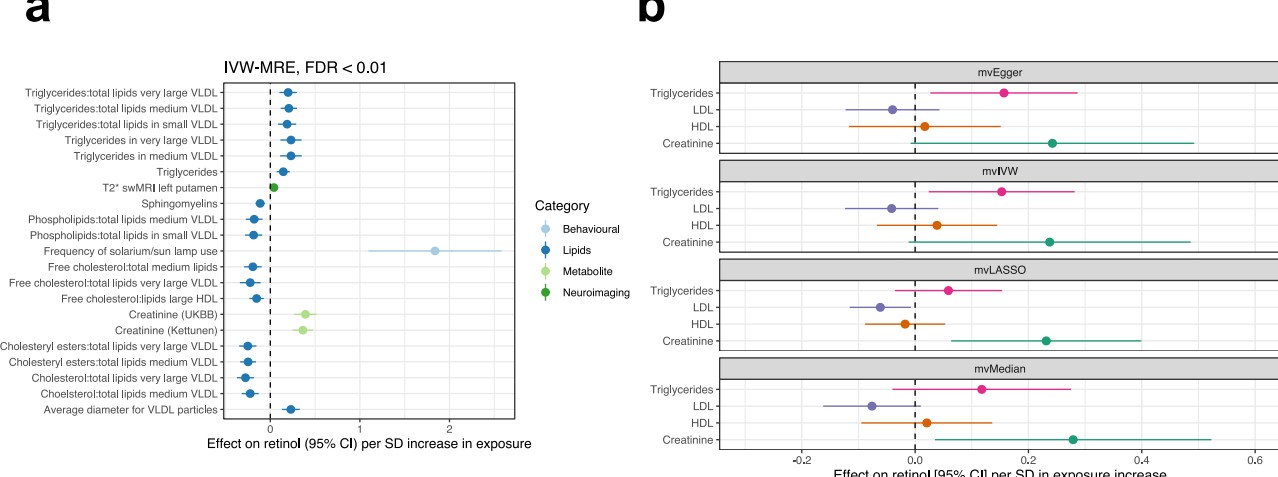

**b**

**Fig. 5 | Exploring the causal effects of continuous exposures on circulating retinol. a** Exposure traits that demonstrated a significant causal estimate (IVW-MRE) on circulating retinol after multiple-testing correction (FDR < 0.01). Traits are coloured relative to their broad phenotypic category. **b** Multivariable MR (MVMR) models investigating the effect of creatinine and major lipid species on circulating retinol. Each panel represents the results from a different MVMR model (each with different underlying assumptions (Online Methods)). The exposure – retinol relationship plotted is conditional on the three other traits in the model.

fixed effects, and Tier #2/Tier #3 traits still exhibit non-zero evidence across multiple methods and no indication of a single IV driving the association. Further, using instead a fixed effects IVW estimator as the primary test for trait pairings with no heterogeneity (Cochran's $Q$ $P < 0.05$) yields similar outcomes being prioritised as most statistically significant after FDR correction (Supplementary Note 2). It is also important to consider when interpreting these estimates that MR approaches are only valid under the assumptions they make, and as a result, any potential causal relationship reported here requires further validation, ideally using a randomised control trial design.

We hypothesised that the putative effect of retinol on body fat could be one explanation for its relationship in this study to brain phenotypes beyond a direct effect of retinoid signaling, particularly given that obesity and adiposity have been linked with brain indices captured by MRI[47]. However, using IVs for body fat percentage, we did not find any strong evidence that it is causally related to any of the retinol-associated brain regions (Supplementary Data 21), suggesting a direct effect of retinol on these regions/networks or a relationship induced through some other unobserved confounder.

We then applied the above pipeline to the FinnGen r8 traits. There were eight disease phenotypes that retinol was associated with after multiple testing corrections (Supplementary Fig. 9, FDR < 0.01), which increased to 19 with an exploratory FDR < 0.1 threshold (Supplementary Data 22). There was a moderate degree of concordance between the FinnGen r8 phenome-wide results using the single *RBP4* IV and the IVW-MRE method ($r = 0.48$). We found three disease endpoints with Tier #3 evidence (Fig. 4B, Supplementary Data 23). Specifically, genetically predicted retinol was estimated to increase the odds of congenital malformations of the heart and great arteries, whilst it was protective for type 2 diabetes with coma and inflammatory liver disease. As these traits had tier #3 evidence, there was some inconsistency in the strength of the results across different MR methods, and therefore, these relationships should be interpreted cautiously. One of the most active areas of research in retinol epidemiology is the relationship between retinol and cancer risk[48]. The estimated effect of circulating retinol on the odds of any malignant neoplasm was not significantly different than one - OR = 0.97 [95% CI: 0.91, 1.04], $P = 0.423$ (Supplementary Fig. 10). However, there was some indication of a protective effect of retinol on squamous non-small cell lung cancer, which approached the threshold for statistical significance after FDR correction - OR = 0.64 [95% CI: 0.51, 0.80], $P = 8.19 \times 10^{-5}$, $q = 0.01$.

There was also some data to support retinol having effects on other respiratory neoplasm endpoints (Supplementary Fig. 10). Using the single *RBP4* IV, there was additionally some weak, nominal evidence for a protective effect of retinol on all non-small cell lung cancers ($P = 2.3 \times 10^{-3}$, $q = 0.45$) Given previous observational evidence that retinol is protective for lung cancer[48], as well as some data supporting the use of synthetic retinoids like bexarotene in lung neoplasms[49], this relationship warrants further exploration.

## Genetic evidence that lipids and kidney function influence circulating retinol

It is also clinically valuable to understand exposures and diseases that impact circulating retinol abundance. To explore this in greater detail, we leveraged retinol as an outcome trait in MR analyses. We utilised a diverse range of thousands of continuous and ordinal phenotypes from IEUGWASdb as exposures in a similar pipeline described above (Online Methods). Several lipid species were demonstrated to putatively influence retinol abundance after multiple-testing correction (FDR < 0.01, Fig. 5A, Supplementary Data 24); for example, triglycerides were implicated to increase circulating retinol whilst cholesteryl ester-related traits decreased circulating retinol. We also saw some evidence for a positive effect of the frequency of solarium and sun lamp use on retinol, which may arise from behavioural-related mechanisms. Furthermore, our findings suggest that increased retinol levels are associated with a susceptibility-weighted MRI measure in the left putamen, called T2*. T2*, reflecting magnetic susceptibility relative to tissue water, can be influenced by factors such as iron and calcium content. This association may also be influenced by behavioural or other pathways that warrant further investigation.

After applying the same tiering system used for retinol as an exposure, we observed Tier #1 evidence strongly supporting a causal effect of creatinine on circulating retinol. This relationship is biologically plausible and may represent an association with kidney function[50,51], although circulating creatinine is biologically heterogenous and is also influenced by unrelated factors like muscle mass. Due to the biological complexity of lipid traits, we observed significant heterogeneity between IV effects, and as a result, they were not assigned a tier in our analysis. However, the effect of triglycerides on increasing circulating retinol levels is consistent with established knowledge of retinol biology, as well as this study implicating two genes that are mechanistically confirmed to impact triglycerides

(*GCKR* and *MLXIPL*). We then leveraged the CAUSE model to distinguish causal effects of creatinine on circulating retinol from correlated pleiotropy that may arise between these two traits due to the extensive polygenicity of creatinine[52]. We found that a model that includes a causal effect of creatinine on retinol was more parsimonious than a model of pleiotropy alone (sharing model) through comparison of these models using the Bayesian expected log pointwise posterior density (ELPD) method (Supplementary Fig. 7, $\triangle\text{ELPD}_{\text{Sharing vs Causal}} = -4.34$, $P = 8.9 \times 10^{-3}$). Given that IVs for creatinine could plausibly act through lipid species like triglycerides to influence circulating retinol, we then constructed multivariable MR (MVMR) models that estimated the creatinine to retinol relationship conditioned on high density lipoprotein (HDL), low density lipoprotein (LDL), and triglycerides (Online Methods). While there was some evidence that the effect of creatinine on retinol could arise due to triglycerides, there was also evidence to suggest an independent effect of both triglycerides and creatinine on increasing circulating retinol, depending on the modelling parameters used (Fig. 5B, Supplementary Note 3).

Finally, we explored pharmacological agents and molecular perturbagens that may influence circulating retinol (Online Methods). Considering the novel genes prioritised in this study with an assigned direction of expression (TWAS/PWAS, pQTL MR), it was found that *GSK3B* is a drug-target known to be inhibited by lithium and related compounds. This may be of clinical interest as it suggests that lithium, utilised as a therapy in mood disorders, may decrease circulating retinol via its inhibition of *GSK3B* given that genetically predicted expression of this gene was positively associated with retinol. We then employed computational signature mapping to further characterise pharmacological agents related to retinol (Online Methods). However, these analyses did not yield any compounds for which the in vitro transcriptomic signature significantly matched or opposed genetically predicted expression associated with retinol after multiple-testing correction (Supplementary Data 25). We then considered perturbagen signatures aggregated to biological pathways or compounds grouped by overall mechanisms of action (MOA) (Online Methods). After multiple-testing correction (FDR < 0.05), there were 13 gene-set based perturbagen signatures that were significantly similar to the directionality of genetically predicted expression associated with retinol (Supplementary Data 26). For example, the expression signature of compounds in the HDAC inhibitor MOA opposed expression genetically predicted to increase serum retinol. This can be interpreted as while no single HDAC inhibitor was significantly associated with retinol, there was at least some evidence for the overall relationship with this MOA. This accords with the suggested effect of HDAC inhibitors like valproic acid on downregulating the expression of *RBP4*[53–55], a gene not included in the signature mapping analyses given the large effect of its encoded protein on retinol.

## No strong evidence for traits which exhibit bidirectional relationships with circulating retinol

We then considered whether any of the tested traits may exhibit a bidirectional causal relationship with circulating retinol. Reverse causality for Tier #2 and Tier #3 traits from the IEUGWASdb pipeline was first assessed, although for binary traits this should be treated purely as a test of the null hypothesis given the difficulties in using IVs for binary traits[56]. There was no strong evidence for reverse causality of any of these traits (Supplementary Data 27). One exception to this was in relation to expression of the protein PEAR1, for which there was very nominal evidence of bidirectional effects. We also investigated evidence for bidirectional effects involving FinnGen Tier #3 traits that retinol is genetically predicted to influence. The use of either genome-wide significant or suggestively significant SNPs ($P < 1 \times 10^{-5}$) as IVs did not indicate evidence of reverse causality of these diseases to retinol (Supplementary Data 28). However, given some of the statistical limitations of these analyses, such effects warrant further consideration.

Lastly, we considered whether creatinine, as the most confident trait pairing with retinol as the outcome, exhibited a bidirectional relationship. There was only very weak evidence that circulating retinol has a negative effect on circulating creatinine ($P = 0.023$).

## Genetically proxied retinol can identify individuals outside of the normative range of circulating retinol for a given age

We were also interested in evaluating the performance of a genetically proxied index of circulating retinol, that is, a circulating retinol polygenic score (PGS). The independent TwinsUK cohort was utilised to tune and evaluate retinol PGS (Online Methods). We used several methods to evaluate the performance of a retinol PGS in this cohort. Firstly, we test different retinol PGS configurations in a random selected training subset of the model (70% of cohort), using a linear mixed model to account for relatedness between the twin pairs. The best performing retinol PGS configuration in the training subset explained approximately 2.12% of the phenotypic variance of retinol when applied to the remaining 30% of the cohort (mean variance explained across three retinol measurement timepoints). We also found similar performance when applied to each subset of twin pairs (Supplementary Data 29). A limitation of this approach for using the same cohort for tuning and testing the retinol PGS is that the estimated effect sizes may not be representative. As a result, we employed an approach to tune weights for the PGS using the summary statistics alone. This was achieved through leveraging the principles of probabilistic finemapping to update variant weights by their posterior probability of association (Online Methods). Due to the modest polygenicity of retinol, this method upweights a small number of variants. Despite this, these scores were still significantly associated with circulating retinol in TwinsUK (Supplementary Data 29).

Like most micronutrients, circulating retinol has been shown previously to have a complex relationship with age[57]. However, population-level approaches investigating these effects do not account for inter-individual variability. We hypothesised that normative modelling could be used to characterise individual patterns of circulating retinol, and to evaluate the contribution of genetics to these individualised effects. Normative modelling, derived from the application of growth charts in paediatric medicine, aims to estimate normative reference ranges of variation in the population (e.g., of circulating retinol) based on age and/or other relevant variables. Here, we established reference ranges for circulating retinol as a function of age using generalised-additive models for location, scale, and shape (GAMLSS) frameworks in the TwinsUK dataset (Online Methods, Supplementary Note 4). These models were constructed in the full cohort, as well as in one twin subset only for comparison. Briefly, this involved identifying the optimal distribution for the GAMLSS model using all samples, followed by splitting the data into two partitions. One partition was utilised to estimate centiles (5th, 25th, 50th, 75th and 95th), and the left-out subjects were benchmarked against this reference chart to determine the position of their individual retinol measurement (Online Methods). This process was then repeated using the opposite subject allocation. An individual's retinol level was classified as infra-normal if their measured retinol fell below the 5th percentile for their age, and as supra-normal if their retinol value exceeded the 95th percentile for their age.

We then investigated the extent to which retinol PGS was associated with these individual profiles of circulating retinol with respect to age (Supplementary Data 30). By way of example, we report results forthwith from the more conservative modelling approach which only used one half of the twins. Retinol PGS was at least nominally significantly associated with supra and infra-normal deviations except for supra-normal deviations at the first (youngest) visit for which there was only a trend observed (Supplementary Figs. 12 13). For example, at the second visit each SD in retinol PGS was associated with an approximately 70% [95% CI: 23%, 136%] increase in the odds of exhibiting

supra-normal retinol levels for a given age relative to all remaining participants, whilst conversely reducing the odds of displaying infra-normal levels by approximately 37% [95% CI: 12%, 55%]. These data suggest that genetics is a non-zero contributor to circulating retinol levels that fall outside the normative range for a given age. In future, a normative modelling approach could be utilised to examine additional factors, including dietary intake, as well as the interplay between genetics and other influences, that might contribute to individualised deviations in retinol levels relative to the population benchmarks.

## Discussion

We conducted the largest GWAS of circulating retinol to-date, revealing important insights into genetic influences on this trait. The sample sizes in this study facilitated the first published estimate of SNP heritability for circulating retinol, plausibly between 5-10%. However, the large standard errors accompanying these estimates reinforces that greater sample sizes are still needed. Moreover, we were able to uncover confident genetic signals associated with retinol at genome-wide significance outside of the RBP4:TTR transport complex. The gene prioritisation pipeline applied both within and beyond genome-wide significant loci prioritised eight genes with high-confidence for a role in retinol biology. These genes were highly expressed in the liver and overrepresented amongst biological pathways including carbohydrate metabolism. The liver is known to be the key organ responsible for retinol storage and processing[1], which is represented strongly by the genetic data in this study. Further, the prioritised genes assigned as overrepresented in the *regulation of carbohydrate metabolic process* pathway (*GCKR*, *GSK3B*, and *MLXIPL1*) are all broadly known to be related to hepatic energy metabolism. As lipids are directly mechanistically linked to retinol absorption, storage, and delivery[1,3], it is plausible that the varied metabolic roles of these genes converge on changes in the abundance of different lipid species. The role of lipids in circulating retinol abundance is also highlighted by our Mendelian randomisation analyses. However, it is still likely that gly-caemic homoeostasis may impact circulating retinol via mechanisms not directly linked to lipid biology; for example, expression of the insulin-controlled glucose transporter GLUT4 is postulated to be related to RBP4 protein levels[58]. In summary, our results suggest that the most identifiable common variant influences on circulating retinol are either mediated through direct effects on transport or metabolic factors, particularly related to lipids. We also prioritised genes like *FOXP2* for which a mechanistic relationship to retinol is less inherently clear. Our analyses of transcriptomic correlates of *FOXP2* supported the immense biological pleiotropy associated with this transcription factor, reinforcing its significance outside of its traditionally conceived association in the literature with neurological phenotypes like language. Work is now needed to disentangle the mechanisms which specifically underlie this relationship between *FOXP2* and circulating retinol that were infer from these genetic findings.

Our study also represents a significant advancement as it is the first to perform a high-throughput, hypothesis free, analysis investigating the potential causal effects of retinol across a wide range of human clinical phenotypes using Mendelian randomisation. This work recapitulated known influences of retinol on ophthalmological measures[59], the innate and adaptive immune response[60], and congenital heart malformations[61]. However, we also uncovered some less characterised relationships that may be of direct clinical relevance. We highlight forthwith the example of circulating retinol being genetically predicted to impact the thickness and surface area of several brain regions, as well as indices of brain connectivity. Retinoic acid, a downstream metabolite of retinol, is considered one of the most intrinsic central nervous system signalling molecules, particularly as it exerts control over processes like neuronal differentiation *in utero* and adult neurogenesis, as reviewed elsewhere[2]. It is, therefore, logical that retinol would plausibly influence brain structure and connectivity

throughout the lifespan. However, the regions implicated in this study require further examination with respect to their clinical significance. By way of example, we associated increased circulating retinol with a reduction in thickness in the right rostral anterior cingulate cortex. This cortical region has been identified by a large international mega-analysis from the ENIGMA consortium to exhibit increased thickness in individuals with the neuropsychiatric disorder schizophrenia compared to controls[62], suggesting a potential protective effect of retinol in this region with respect to schizophrenia. This accords with previous evidence linking retinoids to schizophrenia[2,63]. It is known clinically that both retinol deficiency and toxicity can have harmful neurological effects, highlighting the complexity of the relationship of retinol to the brain throughout the human lifespan. This complexity is also seen with retinoids used as pharmacotherapies. For instance, isotretinoin (13-*cis* retinoic acid), indicated for conditions like acne, has been shown to have opposing effects on adult neurogenesis relative to all-*trans* retinoic acid and putatively increases the risk of suicide[64,65], although evidence for this association is mixed[66]. Conversely, a synthetic retinoid, bexarotene, with different receptor affinities to isotretinoin, has demonstrated some promise as a potential adjuvant to antipsychotics in schizophrenia[67]. Future work should attempt to understand these relationships with greater fidelity by investigating genetic influences on other retinoids beyond retinol, as well as how tissue-specific abundance can differ from what circulates in serum[68]. Emerging methods for non-linear Mendelian randomisation would also be useful in this context given that retinol often exerts dose dependent effects[69]. The causal estimates generated in this study also need to be treated with appropriate caution due to the limitations of MR and require further validation in study designs that can enable causal inference, such as randomised control trials. As reviewed previously[18,19,70], MR tests are only unbiased when their assumptions are plausibly satisfied, which is why we implemented a suite of different methods and sensitivity analyses with quite distinct underlying assumptions in this study. Genetic estimates on circulating retinol used as IVs could also be confounded by factors including uncontrolled population stratification, selection bias, and measurement error. We also can compare our MR-pheWAS results to some previous literature that used retinol IVs in a MR study, most of which used *RBP4* and *TTR* IVs from the previous Mondul et al. GWAS. For example, we replicated a largely null effect of circulating retinol that was previously suggested using MR on the odds of Alzheimer's disease[71], digestive tract cancers[72,73], and endometrial cancer[74]. Interestingly, a previous MR study did report weak evidence of a risk increasing relationship between genetically proxied dietary intake of retinol and lung cancer, which was the opposite direction of that we found for measured circulating retinol in this study[75]. However, MR studies of dietary retinol intake are very difficult to interpret due to the pleiotropic nature of any genetic effects on nutrient intake patterns that would be used as IVs, highlighting the importance of GWAS using measured retinol. In summary, we provide a large resource to the literature of putative effects of circulating retinol across the human clinical phenome that will be informative for future investigation of this trait.

Finally, we make some recommendations for future GWAS of retinoid molecules. An important limitation of our analyses is that we only investigated genetic effects on circulating retinol, rather than retinol availability within target tissues. Given the complexities of retinol homoeostasis, it is plausible that genetic effects on factors like retinol entry into the cell (e.g., STRA6 receptor) and esterification for storage (e.g., lecithin retinol acyltransferase) are obscured when considering only retinol present in serum or plasma. Therefore, future studies should attempt to measure retinol abundance in different tissues from genotyped samples, although this will pose a challenge in terms of obtaining sufficient sample sizes. Moreover, it would be of interest to characterise the genetic overlap between effects on retinol versus other retinoids like retinaldehyde and all-*trans* retinoic acid.

Despite these limitations, and the need for concerted efforts to collect more data, we believe this study demonstrates the value in conducting retinol GWAS to both better characterise retinoid associated biology and its clinical significance.

## Methods
### Study cohorts
The study was designed and conducted in compliance with all the relevant regulations regarding the use of human study participants and the criteria set by the Declaration of Helsinki. Each contributing study cohort to the meta-analysis was approved for data collection and use by their individual institutional review boards (INTERVAL: all participants gave informed consent, National Research Ethics Service approved the study (11/EE/0538); METSIM: all participants provided written informed consent, with approval by the ethics committee at the University of Eastern Finland and the Institutional Review Board at the University of Michigan; ATBC/PLCO: IRB approval by NIH). The TwinsUK Resource Executive Committee (project ID: E1205) approved the use of the TwinsUK dataset and the related study protocol. The use of the TwinsUK data complies with the ethics regulations of the TwinsUK Resource Executive committee. The submitted manuscript and protocol was reviewed by the TwinsUK Resource Executive committee and internally by NIH. The proceeding section outlines the datasets included in the genome-wide meta-analysis of circulating retinol, as well as the replication cohort.

**INTERVAL.** The largest constituent cohort of the meta-analysis was drawn from the INTERVAL study, comprised of recruited blood donors from the United Kingdom[76]. Retinol abundance was measured from plasma using the high-throughput metabolomics platform DiscoveryHD4® (Metabolon, Inc., Durham, USA), as outlined in Supplementary Note 5. Briefly, after adjustment for various technical/biological confounders and outlier effects, residualised plasma retinol was inverse-rank normal transformed before association testing. Whole-genome sequencing of this cohort was performed as described elsewhere[77]. A GWAS of the normalised residuals was performed in HAIL via multiple-linear regression adjusted for INTERVAL metabolon batch and 10 genetic PCs. The final GWAS sample size was 11,132 European ancestry participants.

**METSIM.** Plasma retinol was also measured using the DiscoveryHD4® high-throughput platform, the same platform as INTERVAL, in a recent metabolome-wide GWAS of the METSIM (Metabolic Syndrome in Men) study[78]. Analogous to INTERVAL, METSIM was a cohort of recruited volunteers. The cohort consists of middle-aged men recruited from Northern Finland between 2005-2010[79]. As described by Yin et al.[78], METSIM participants were genotyped using the Human OmniExpress-12v1_C BeadChip and imputed using a custom METSIM panel of whole genome-sequenced participants in the study. A linear mixed model implemented in EPACTS v.3.2.6 was then leveraged to perform GWAS on residualised retinol, subjected to inverse-rank transformation after adjustment for technical and biological confounders. The final GWAS had a sample size of 6136 METSIM participants (European ancestry).

**ATBC+PLCO.** The largest previous dedicated GWAS of circulating retinol from 2011 was also included in this study[22]. This GWAS comprised data from two studies that measured serum retinol: the Alpha-Tocopherol, Beta-Carotene Cancer Prevention (ATBC) Study, a Finnish randomised control trial of beta-carotene/alpha-tocopherol supplementation for cancer prevention[80], and the Prostate, Lung, Colorectal, and Ovarian (PLCO) Cancer Screening Trial, a United States trial of cancer screening effectiveness[81]. The inclusion criteria, measurement of serum retinol, and genotyping have been outlined by Mondul et al.[22]. Briefly, ATBC/PLCO samples were genotyped using the Illumina

HumanHap550/610 arrays and imputed to the HapMap Central European reference panel, whilst serum retinol concentrations were estimated using reversed-phase liquid chromatography. The retinol GWAS (N = 5006) was performed in R (version 2.10.1) using multiple linear regression adjusted for age at sample collection, SNP derived PCs, cancer status, serum cholesterol, and body mass index. The retinol units for the GWAS effect sizes were in natural log transformed μg/L. A limitation of using summary statistics from this study is that only <600,000 variants were available for inclusion in the GWAS, as was common at the time before more recent advances in imputation pipelines that result in larger post-imputation yield. Therefore, to increase the number of variants available for meta-analysis, we applied a summary statistics-based imputation procedure to boost the number of variants available for meta-analysis. After harmonisation with the 1000 genomes phase 3 reference panel, we applied Gaussian summary statistics imputation (ImpG) as implemented by the FIZI v0.7.2 python package (https://github.com/bogdanlab/fizi) with the default window size of 250 kb[82]. The ImpG model leverages the assumed Gaussian distribution of GWAS $Z$ scores with a mean of zero and variance that arises from the LD-induced correlation between variants. As outlined elsewhere[82], $Z$ scores of unobserved (imputed) variants can be estimated given the LD correlation matrix derived from the reference panel, with a metric of imputation accuracy ($R^2$) calculated using the conditional variance. We retained only confidently imputed variants ($R^2 > 0.8$).

**TwinsUK.** We performed a serum retinol GWAS in the TwinsUK cohort to serve as a replication dataset. High-throughput metabolomics profiling in this cohort has been described extensively elsewhere[83]. TwinsUK is a prospective population-based study of mostly female twin pairs which has been profiled using a variety of multiomic technologies[84]. As outlined by Shin et al., genotyping was performed with a combination of Illumina arrays (HumanHap300, HumanHap610Q, 1M-Duo and 1.2MDuo 1 M)[85]. This was followed by imputation to the 1000 genomes phase reference panel after quality control and retaining individuals of predominantly European ancestry after PCA. We retained physically genotyped variants and those with at least moderate imputation accuracy for GWAS ($R^2 > 0.3$) in 5654 samples. Our strategy for GWAS in this cohort given the limited overlap between participants with both measured retinol and genetics available was to split the twin pairs into two separate cohorts and ensure individuals in each sub-cohort were unrelated. Relatedness testing in both sub-cohorts containing one of the two possible twin pairs was performed separately using KING as implemented by plink2 (PLINK v2.00a3LM AVX2 Intel), with one participant from third-degree relative or greater pairs randomly removed[86,87]. Kinship estimation via KING was performed for autosomal variants that satisfied all of the following: physically genotyped on the array, MAF > 0.05, outside of regions of long-range LD like the MHC[88], and in relative linkage equilibrium ($r^2 < 0.05$). PCA was then applied in each sub-cohort using plink2 to calculate eigenvectors for use as downstream covariates. Retinol was measured from samples at three timepoints. The mean age of participants at each timepoint was 51.5 (SD = 8.41), 58.6 (SD = 8.38), and 64.7 (SD = 8.41), respectively. There were only a very small number of males in this cohort (~ 3%), so only females were retained for further analysis due to this imbalance in sex composition. After merging with the genotyped split twin cohorts, as described above, there were up to 918 and 717 genotyped participants with measured retinol at three timepoints in each subset, respectively. Six GWAS were performed: in each sub-cohort (one or two), multiple linear regression was utilised to test the additive effect of each variant on measured retinol at one of the three measured time-points covaried for age, five SNP derived PCs, and metabolomics batch. These GWAS was performed using the --glm flag in plink2, resulting a 9,051,192 by three matrix of estimated additive genetic retinol effect sizes for both sub-cohorts.

## Genome-wide meta-analyses

We conducted genome-wide meta-analysis of common variants (MAF ≥ 0.01) using METAL (version March 2011), followed by a rare variant (MAF < 0.01) meta-analysis also with METAL. The METSIM and INTERVAL cohorts were integrated for the primary meta-analysis as they both had expansive genome-wide coverage of common and rare variants. A sample size weighted meta-analysis of $Z$ scores (Stouffer's method) was utilised for this purpose. We also conducted the MET-SIM + INTERVAL meta-analysis via an inverse-variance weighted estimator with fixed effects to estimate the effect sizes of effect alleles in SD units of plasma retinol given this was the unit of both the METSIM and INTERVAL GWAS, as well as both studies using the same Meta-bolon Inc. platform for metabolite quantification. Heterogeneity between the studies was assessed using Cochran's $Q$ test. We then conducted a meta-analysis with fewer available variants that also included ATBC + PLCO using Stouffer's method (as the unit for this GWAS differed from METSIM and INTERVAL). This larger sample-size meta-analysis was also restricted to common variants as there was very limited rare variant coverage in ATBC + PLCO.

The FUMA v1.4.1 (Functional Mapping and Annotation of Genome-Wide Association Studies) platform was utilised to annotate variants, define lead SNPs, and infer loci boundaries for genome-wide significant signals ($P < 5 \times 10^{-8}$)[89]. We utilised the default settings for defining independent significant SNPs ($r^2 \leq 0.6$) and lead SNPs ($r^2 \leq 0.1$). LD estimation was achieved using the 1000 genomes phase 3 European reference panel, with LD blocks within 250 kb of each other merged into a single locus. We then attempted to replicate the lead SNPs from the eight genome-wide significant loci (METSIM + INTERVAL) in TwinsUK, as described in the previous section. Given the small sample size of TwinsUK, we sought to ascertain if the lead SNPs were directionally consistent with the meta-analysis. This was achieved by taking the mean SNP $Z$ score across the three timepoints for the two sub-cohorts of unrelated participants, with a binomial test utilised to infer whether the number of lead SNPs (mean $Z$) that were directionally consistent was greater than chance alone (Binomial $P < 0.05$).

In the rare-variant meta-analysis, we annotated variants using the Functional Annotation of Variants (FAVOR) online resource[90]. Phenome-wide association profiles of selected variants were also investigated using the pheweb browser collated from FinnGen release 8 (https://r8.finngen.fi/)[91]. Rare variants were then aggregated to genes through leveraging the characteristics of the Cauchy distribution[92,93]. In this approach, gene-wise $P$ values are summed and then transformed to approximate a Cauchy distribution, which due to its heavy tail is insensitive to correlations amongst the $P$ values. This behaviour of the Cauchy distribution is important as covariance amongst rare variants is difficult to estimate, and therefore, this approach guards against inflated type I error due to potential unknown covariance/LD between rare variants. Code for implementing the Cauchy aggregation was adapted from https://github.com/yaowuliu/ACAT.

## SNP heritability estimation

Summary statistics for the meta-analyses were munged using the munge_sumstats.py script from the *ldsc* repository of scripts (https://github.com/bulik/ldsc) and only common (MAF > 0.05) HapMap3 variants outside of the MHC retained. SNP heritability was then estimated using the LDSR model and the 1000 genomes phase 3 reference panel[94]. By way of comparison, we then estimated SNP heritability using the LDAK model via SumHer as implemented in LDAK v.5.2 (https://dougspeed.com/)[95]. Pre-computed tagging files derived from 2000 white British individuals in the UKBB for HapMap3 SNPs were utilised to calculate SNP heritability using the LDAK-thin and BLD-LDAK models[95,96]. Briefly, the same munged summary statistics were used for these two LDAK heritability models. In these two approaches, the SNP-wise expected heritability is not fixed but rather it is free to vary by MAF and local LD in the LDAK-thin model, whilst the BLD-LDAK

model additionally varies this by functional annotation. We also estimated partitioned SNP heritability via LDSR using a multi-tissue and cell-type panel[97].

## Empirical Bayes' modelling of the genetic architecture of retinol

We investigated the polygenicity of the genetic architecture of circulating retinol using an Empirical Bayes' adaptive shrinkage method termed ashR[98]. Functions to perform this method were implemented via the ashR R package v2.2-54 (https://github.com/stephens999/ashr). Briefly, this approach models effect sizes, along with their standard error, as a mixture of zero and non-zero effects. Empirical Bayes' inference is performed under the assumption that the distribution of these variant effects is unimodal, which is a realistic assumption for genetic effects on complex traits at the population level. The MET-SIM + INTERVAL IVW meta-analysis was utilised for this as it provides an interpretable effect size and standard error for each variant (plasma SD units). In line with previous work in the literature[99], we annotated each HapMap3 variant with its corresponding LD score from the 1000 genomes phase 3 European reference panel, as well LD scores from the UKBB White Great British participants for comparison, and sorted these into bins of similar LD scores ($N_{Bins}$=1000 and $N_{Bins}$ = 5000). The ashR Empirical Bayes' inference of the proportion of non-zero effects was undertaken in each LD score bin, followed by calculating the mean across all bins. We utilised a generalised additive model to plot a smoothed trend line of the relationship between increasing LD score bin (higher LD score) and the proportion of non-zero effects.

## Gene prioritisation

Gene prioritisation was performed within genome-wide significant loci, as well as outside of loci that obtained genome-wide significance. Due to the better coverage of common variants, we focused on the METSIM + INTERVAL meta-analysis for gene prioritisation. The pipeline for prioritising putative causal genes within the eight genome-wide significant loci was adapted from a previous GWAS performed by our group[100]. The following criteria were utilised in this study: 1) closest transcription start site (TSS) to the lead SNP, 2) closest gene (any) to the lead SNP, 3) gene encoding retinoid transporter or enzyme in locus, 4) non-synonymous variant in locus, 5) the most statistically significant GTEx eGene [expression quantitative trait loci (eQTL) signal] in the locus, 6) most significant GTEx eGene in finemapped credible set for eQTL signal [posterior inclusion probability ($PIP$) > 0.1, DAP-G method – a method for finemapping likely causal eQTL signals from non-causal associations with gene expression][101,102], 7) strongest plasma pGene [protein quantitative trait loci (pQTL) signal] drawn from finemapped pQTLs ($PIP$ > 0.5)[103], 8) highest CADD score, which is a ranking metric of the deleteriousness of a particular allelic substitution relative to the rest of the genome; for example, a CADD score > 20 represents the predicted top percentile of the most deleterious substitutions[104], 9) lowest RegulomeDB score, which integrates eQTL and epigenetic data to predict the most likely functional variants in the non-coding genome[105], 10) the OpenTargets *V2G* predicted gene for the lead SNP[106], and 11) genes physically mapped to SNPs in the 95% credible set derived from probabilistic finemapping (assuming a single causal variant such that LD did not have to be modelled)[107]. We used a prior variance of 0.15 to approximate Bayes' factors from variant-wise effect sizes (METSIM + INTERVAL IVW meta-analysis), in line with previous work finemapping association signals with quantitative traits[108]. We characterised which genes satisfied the greatest number of the above criteria on a per locus basis.

*FOXP2* was one of our confidently prioritised genes but its biological significance outside of the brain is less well understood. To investigate this, we analysed RNA sequencing data from an in vitro experiment that overexpressed *FOXP2* in a human osteosarcoma epithelial cell line (U2OS) via transfection of wild-type *FOXP2* expressing plasmids. Raw read counts from five control cell line replicates versus

five plasmid transfected *FOXP2* overexpression replicates were downloaded from the Gene Expression Omnibus (GEO) resource (GEO Accession: GSE138938)[32]. Data normalisation, filtration, and differential expression analyses were performed using the edgeR package version 3.34.0[109]. Specifically, raw counts were firstly normalised to library size and lowly expressed genes with fewer than 10 raw counts in the smallest library were removed via a counts-per-million thresholding approach. Data were inspected before and after the filtration step via coefficient of variation (BCV) and multidimensional scaling (MDS) plots (Supplementary Fig. 14). Differential expression for each gene that survived quality control was then performed using exact tests for differences in the means between two groups of negative-binomially distributed counts. We defined a differentially expressed gene as those which survived multiple-testing correction using the Bonferroni method ($P_{Corrected} < 0.05$), with three different absolute $\log_2$ fold change (FC) cut-offs considered: $|\log_2 FC| > 1.5$, $|\log_2 FC| > 2$, and $|\log_2 FC| > 5$. The use of Bonferroni correction and large absolute FC thresholds is very conservative; however, given the volume of differentially expressed genes, and a relatively large number of replicates for a cell line experiment boosting power, we believe these strict parameters are warranted to prioritise the most salient *FOXP2* associated signals. The overrepresentation of each set of candidate genes amongst biological pathways and other ontology sets was tested using g:Profiler[110].

To identify potential causal genes that have not reached genome-wide significance at our current sample size, we integrated the circulating retinol GWAS with genetic effects on mRNA and protein expression. Firstly, we conducted a transcriptome and proteome-wide association study (TWAS/PWAS) of circulating retinol using the FUSION approach[111]. As outlined previously[112–115], FUSION leverages models of *cis*-acting genetically regulated expression (GReX) that exhibit statistically significant non-zero heritability. Variant weights from GReX models are integrated with the effect of those same variants on retinol to estimate the direction of genetically regulated expression associated with increasing circulating retinol. TWAS GReX (mRNA) were estimated previously using GTEx v8 (http://gusevlab.org/projects/fusion/). We selected the following biologically informative tissues to perform TWAS based on known retinol biology or tissues that exhibited at least nominally significant ($P < 0.01$) enrichment of SNP heritability in the partitioned-LDSR model. The selected tissues were: small intestine terminal ileum, pancreas, liver, adipose (visceral omentum), adipose (subcutaneous), breast (mammary tissue), and whole blood. It has been suggested previously that TWAS signals from tissues that are less directly trait relevant can induce spurious associations, which is why we limited our hypothesis space to these tissues[116]. Protein GReX were derived from plasma (ARIC study, European subset), as outlined elsewhere[103]. We applied Benjamini-Hochberg false discovery rate (FDR) correction across all TWAS $Z$, followed by all PWAS $Z$. Colocalisation between GReX models and retinol was performed for all TWAS/PWAS signals that were at least nominally significant via the coloc package as implemented by FUSION (single shared variant hypothesis)[108]. We then considered a more conservative approach to prioritise proteins for whom expression could be causally linked to circulating retinol through leveraging finemapped plasma pQTLs ($PIP > 0.5$, ARIC, European subset) as instrumental variables (IV) for Mendelian randomisation[19,45,103]. The proteomic platform used was the SomaLogic Inc. version 4 platform via an aptamer (SOMAmer) based approach. In the ARIC study, 4697 unique proteins or protein complexes were measured. In our study, we focused on finemapped pQTLs within 500 kb of the transcription start site for use as IVs. After harmonisation with the retinol GWAS, 937 unique proteins or protein complexes were able to be tested. The Wald ratio method was implemented for proteins with single IVs, whilst an inverse-variance weighted estimator with fixed effects was utilised for proteins with more than 1 IV. Fixed effects were used for the IVW rather

than multiplicative-random effects in this instance as no protein had > 4 IVs. The use of IV based approaches for identifying trait-associated genes versus GReX has been discussed extensively elsewhere[114]. Colocalisation was also performed for proteins that survived FDR correction. Mendelian randomisation was performed using the Two-SampleMR package v0.5.6, with IVs clumped through leveraging LD from the 1000 genomes phase 3 European reference panel to retain only independent pQTLs ($r^2 < 0.001$).

Finally, we investigated the tissue specificity of prioritised genes from GWAS loci and the TWAS/PWAS/MR approach using FUMA. To do this, we compared the expression of these prioritised genes using a *t*-test (one-sided and two-sided) against all other available genes in GTEx v8 on a per tissue basis (54 tissues), followed by applying Bonferroni correction to these *P*-values[89]. We also conducted pathway analyses of these genes using g:Profiler with default parameters[110].

## Causal inference

We developed and implemented a comprehensive pipeline to leverage this retinol GWAS to identify putative causal effects of retinol on traits across the human clinical phenome, as well as traits that causally influence retinol in the reverse direction. This was achieved using Mendelian randomisation (MR), which has been reviewed extensively elsewhere[19,46,70]. Firstly, we considered circulating retinol as the MR exposure. The lead SNP in *RBP4* was first chosen as a single IV to proxy circulating retinol as out of all the genes implicated in genome-wide significant loci, *RBP4* has exhibits the most specificity in terms of its relationship with serum retinol. We utilised the Wald ratio method to estimate the effect of the circulating retinol increasing rs10882283-A allele (METSIM + INTERVAL IVW effect size) on over 19,000 outcomes in the IEUGWASdb v6.9.2 (accessed on the 16th of December, 2022) resource via the ieugwasr package version 0.1.5[26]. This resource reports for SNPs genome-wide their effect size and statistical significance in terms of association with thousands of collated GWAS from varying sources, including the UK Biobank, GWAS catalog, large GWAS consortia, eQTL and pQTL studies, metabolomics studies, and imaging studies, amongst others. To correct for multiple testing, we applied the false discovery rate (FDR) method, and we retained only those retinol/outcome pairs with estimates that were significant below the 1% FDR threshold ($q < 0.01$). Subsequently, we investigated whether significant trait pairs colocalised using *coloc* v5.1.0 with default priors.

Although the single IV approach is more conservative, power to detect causal effects can be boosted by using all available independent ($r^2 < 0.001$, 1000 genomes phase 3 European reference panel) genome-wide significant SNPs as IVs. The inverse-variance weighted estimator with multiplicative random effects (IVW-MRE) was utilised to estimate the effect of retinol on outcomes from IEUGWASdb for which at least 6 IVs were available in that GWAS[117]. The IVW approach has a zero percent breakdown level as it assumes all IVs are valid for use in Mendelian randomisation, which is often unrealistic in practice. We developed pipeline to prioritise the most reliable causal estimates from the IVW-MRE that survived multiple-testing correction ($q < 0.01$). This involved identifying retinol MR estimates on traits for which the following applied: i) no significant heterogeneity ($P < 0.05$) between IV exposure-outcome effects tested with Cochran's $Q$[118], ii) a non-significant intercept of an MR-Egger model that does not constrain the intercept to pass through the origin[119], and iii) no evidence that leaving out any single IV ablates the statistical significance (at least $P < 0.05$) of the estimate[120]. Traits satisfying these criteria were then assigned a tier (Tier #1, Tier #2, Tier #3) based on the raw statistical significance of the retinol causal estimate using other MR methods besides the IVW-MRE that have different assumptions regarding IV validity. These were: the IVW with fixed effects (does not model heterogeneity like with MRE), the weighted median method[121], the weighted mode method[122], and the

MR-Egger method[119]. The underlying assumptions and methodological considerations of using these methods have discussed extensively elsewhere[123,124]. Traits for which the effect of retinol was at least nominally statistically significant using all five methods were assigned as Tier #1, whilst four tests being statistically significant was Tier #2, and three tests being statistically significant was Tier #3. The $F$-statistic and $I^2$ of the IVs was also assessed to ensure they were well-powered ($F > 10$) and suited for MR-Egger ($I^2 > 0.9$), respectively[125]. Any traits assigned a tier from IEUGWAS were then manually curated to ensure the GWAS was European ancestry and well-powered for the trait of interest. We excluded the FinnGen release 5 outcomes from IEUGWAS at this stage of tiering given we examined these further using release 8 in the proceeding analyses. To investigate the effect of body fat percentage on the retinol-associated MRI indices, we used IVs from a non-overlapping GWAS of that trait (N = 65,831)[126]. We then repeated the above process of binary outcomes with at least 1000 cases from FinnGen release 8, which is not included in the current version of IEUGWASdb at time of analysis, to increase power to detect effects on disease endpoints.

We also systematically investigated continuous outcomes that may causally impact circulating retinol. Continuous outcomes were our focus as causal estimates from binary exposures are difficult to interpret and are often less powered in the context of MR[56]. Exposures were filtered from phenotypes available in IEUGWASdb to retain continuous traits, with further filtering to identify traits with ≥ 5 genome-wide significant, independent ($r^2 < 0.001$, 1000 genomes phase 3 European reference panel) IVs available in the retinol GWAS. The same pipeline as above was then applied (IVW-MRE FDR < 0.01, followed by sensitivity analyses and tier assignment). The causal estimate of creatinine on retinol was a Tier #1 trait, and due to the pervasive polygenicity of creatinine afforded by its large sample size, we followed up this relationship using the MR model "Causal Analysis Using Summary Effect Estimates" (CAUSE), using the CAUSE R package v1.2.0[52]. Briefly, this method is a more polygenic approach and seeks to distinguish casual effects from correlated pleiotropy by fitting competing models that account for these terms and comparing them using ELPD. After using 1 million random variants to estimate nuisance parameters, LD clumping and thresholding was applied to the serum creatinine summary statistics (IEUGWASdb trait ID: met-d-creatinine) in line with the original CAUSE publication ($P < 0.001$, $r^2 < 0.01$, 1000 genomes phase 3 European reference panel). The competing CAUSE models were then fit (null, sharing, and causal), ensuring that all Pareto $k$ estimates were <0.5 during the model comparison using ELPD.

We then performed a multivariable MR (MVMR) analysis to estimate causal effects of creatinine on retinol conditioned on three major lipid species using GWAS from the global lipids genetics consortium (LDL, HDL, and triglycerides)[127]. MVMR was undertaken in accordance with previous work using the R packages MVMR v0.3 and MendelianRandomization v0.6.0[123]. Briefly, this entailed identifying variants associated at genome-wide significance with at least one of the four exposures that are independent ($r^2 < 0.001$), calculating a conditional $F$-statistic for multivariable instruments[128], and applying four MVMR models (IVW, Egger regression, Weighted Median, and a LASSO based penalised regression approach for selecting the optimal IV configuration)[129].

## Drugs and perturbagens associated with circulating retinol

We also considered drugs that may influence circulating retinol. We searched genes prioritised from our pipeline with an assigned direction of retinol-associated expression using DGIdb v4.2.0 and DrugBank v5 to identify retinol-associated genes targeted by drugs[130,131], outside of known drugs that target RBP4 and TTR. Retained drug-gene interactions were restricted to those with known mechanism of action and > 2 lines of supporting evidence. We also utilised computational signature mapping to identify pharmacological agents that may enhance or inhibit

the expression of genes associated with circulating retinol. To boost power for signature mapping, we considered all genes that were nominally associated with retinol in the TWAS/PWAS ($P < 0.05$) that exhibited moderate colocalisation of a shared causal variant ($PP_{H4} > 0.4$). These genes were uploaded to the Connectivity Map Query online tool to quantify the similarity, termed connectivity, with drug perturbagen-associated expression profiles, as outlined in detail elsewhere[132].

## Polygenic scoring

We used the independent TwinsUK replication cohort, described in a preceding section, to investigate retinol polygenic scores (PGS). PGS were applied to genotyped variants and high confidence imputed variants ($R^2 > 0.8$) in TwinsUK. There were two different methodologies implemented to construct PGS: LD clumping and thresholding (LD C + T) and a probabilistic finemapping-based method that scales variant effect sizes based on their posterior probability of causality, thereby upweighting signals more likely to be causal (RápidoPGS)[133,134]. In the LD C + T approach, variants were clumped using the within sample LD of TwinsUK at the following $P$-value thresholds: $5 \times 10^{-8}$, $1 \times 10^{-5}$, $1 \times 10^{-3}$, 0.01, 0.05, 0.1, 0.5, and 1. Additive PGS were then profiled using PRSice2 v2.3.5[135]. The RápidoPGS applies probabilistic finemapping (with Bayes' factors approximated using Wakefield's method) to independent LD blocks genome-wide such that variant-wise posterior probabilities of causality can be estimated. This in essence is a shrinkage approach to PGS to account for double-counting effects that arise due to correlated effect sizes induced by LD; however, does not inherently require an independent genotyped sample from the GWAS for tuning. Approximate Bayes' factors in each LD block were derived assuming a prior variance of 0.15, conventionally used for quantitative traits. This parameter choice was compared to a data-driven approach to estimating the prior variance based on SNP heritability, as outlined elsewhere[134]. Variant-wise effect sizes are multiplied by their posterior probabilities before PGS calculation, as above.

As we are using a twin cohort, there were two different approaches we utilized to tune the optimal PGS configuration from the LD C + T approach. Firstly, we randomly split the cohort into a training (70% of participants) and test (30% of participants) partition. A linear mixed model was then fit in the training partition with fixed effects of PGS, age, and metabolomics batch, as well as a random effect of family ID to account for twin relatedness. This was applied for retinol measured at each of the three visits for all the $P$-value thresholds. The marginal $R^2$ from a null model with no PGS was subtracted from the full model to infer the best-performing PGS in the training partition (mean marginal $R^2$ across three visits). The variance explained of the best performing $P$-value threshold was then estimated using the same approach in the test partition. By way of comparison, we also split the twins into separate unrelated cohorts and used one set as training and one as testing. Fixed effects instead of mixed effects linear regression was then implemented in a similar fashion to above, additionally covaried for five SNP-derived principal components. We then repeated all the above for the two probabilistic finemapping weighted PGS (prior variance = 0.15 and data driven prior variance).

## Normative modelling

We built a normative model of retinol as a function of age per study visit in TwinsUK. This was achieved using a generalised additive model for location ($\mu$), scale ($\sigma$), and shape (GAMLSS)[136,137], implemented in R v4.4.1. The GAMLSS approach is useful in this application as it is semi-parametric and able to account for factors such as heteroskedasticity and non-Gaussian distributions. Our modelling approach can be summarised as follows: we firstly fit a GAMLSS model in the full sample for a variety of GAMLSS distribution families implemented by the *gamlss* R package v5.4.12. The model on retinol at visit $i, i \epsilon \{1,2,3\}$ for each of the GAMLSS families set the $\mu$ term as the first order fractional polynomial of age, along with metabolomics batch as an additional

covariate, with the term $\sigma$ just the first order fractional polynomial of age to model the scale of the distribution. The model fit of each of the tested families was then assessed using the Bayesian information criterion (BIC) and the Akaike information criterion (AIC). We repeated the above also including measured body mass index (BMI) at that timepoint in the $\mu$ term by way of comparison. We chose the GAMLSS family (Box-Cox $t$ distribution) through considering the model performance (minimum AIC and BIC) over all three visits (Supplementary Note 4, Supplementary Figs. 15 and 16). We then split the cohort in half, separating the twins, and fit normative centile curves using the selected GAMLSS family to one half of each batch of retinol measurement, and computed deviations on the other independent sample half on a per batch basis. We repeated this process with the modelling and deviation subsets reversed to compute deviations for all samples. Individuals whose measured retinol was above the model derived 95th percentile were classified as having supra-normal retinol for their age, while those below the 5th percentile were classified as having infra-normal retinol. To guard against overfitting due to the relatedness of twins between the two subsets, we then performed all of the above just using one half of the twins as the full cohort for model fitting, followed by splitting this subset as described above. We tested the relationship between scaled retinol PGS (mean = 0, SD = 1) and infra-normal retinol at each visit was tested in half of the cohort (unrelated) using binomial logistic regression additionally covaried for five SNP-derived PCs. The same models were also constructed for supra-normal individuals.

### Software and operating systems
The primary analyses in this manuscript were performed either on a MacBook Pro (OS X: Ventura 13.3), an in-house linux cluster (Ubuntu 18.04.5 LTS), or the High-Performance Computing Research Compute Grid of the University of Newcastle [Red Hat Enterprise Linux release 8.1 (Ootpa)]. The primary R version utilised was version 4.1.1 (2021-08-10), with some additional analyses using R version 4.0.3 (2020-10-10) (linux cluster). The Python version utilised was either Python 2.7.17 or Python 3.6.9, depending on the requirements of the analyses.

### Reporting summary
Further information on research design is available in the Nature Portfolio Reporting Summary linked to this article.

## Data availability
The full retinol GWAS summary statistics generated in this study are available at https://doi.org/10.5281/zenodo.7905523. The RNAseq data analysed by this study are available in the Gene Expression Omnibus (GEO) under accession code GSE138938. The 1000 Genome Project data utilised for LD calculation can be available at ISGR (https://www.internationalgenome.org/data). The datasets used annotation of variants as eQTLs and pQTLs are publicly available from GTEx (https://www.gtexportal.org/home/downloads) and the Chatterjee lab repository (https://nilanjanchatterjeelab.org/pwas/), respectively. Weights used for the estimation of TWAS test statistics are available from the Gusev lab repository (http://gusevlab.org/projects/fusion/). FinnGen GWAS summary statistics utilised in this study are publicly available from the FinnGen website (https://www.finngen.fi/en/access_results). GWAS data from IEUGWASdb utilised in the study for the MR-pheWAS can be publicly accessed from their website (https://gwas.mrcieu.ac.uk/). LD tagging files for the UKBB utilised for heritability estimation can be sourced from the LDAK website (https://dougspeed.com/ldak/). Metabolite raw relative abundances are available for INTERVAL, a cohort included in the GWAS meta-analysis, at https://www.ebi.ac.uk/metabolights/ (project codes: MTBLS833 and MTBLS834). The TwinsUK data used in this study are available under restricted access to protect participant privacy as outlined by the study protocol of TwinsUK, access can be obtained by approved, bona fide researchers by following the steps detailed by the TwinsUK website: https://

twinsuk.ac.uk/resources-for-researchers/our-data/). Researchers wishing to access TwinsUK data must read the data access policy (https://twinsuk.ac.uk/wp-content/uploads/2022/12/DTR_DataAccessPolicy_2022V1.pdf) and complete a "Data Access Proposal Form" for consideration by the TwinsUK Research Executive Committee. Specific enquires related to data access can be directed to victoria.vazquez@kcl.ac.uk.

## Code availability
Code used in this study is freely available at the following GitHub repository - https://github.com/Williamreay/Retinol_GWAS_code.

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

## Acknowledgements

TwinsUK is funded by the Wellcome Trust, Medical Research Council, Versus Arthritis, European Union Horizon 2020, Chronic Disease Research Foundation (CDRF), Zoe Ltd, the National Institute for Health and Care Research (NIHR) Clinical Research Network (CRN) and Biomedical Research Centre based at Guy's and St Thomas' NHS Foundation Trust in partnership with King's College London. P.S. was supported by a Rutherford Fund Fellowship from the Medical Research Council (grant no. MR/S003746/1). Participants in the INTERVAL trial were recruited with the active collaboration of NHS Blood and Transplant England (www.nhsbt.nhs.uk), which has supported field work and other elements of the trial. DNA extraction and genotyping were co-funded by the National Institute for Health and Care Research (NIHR), the NIHR BioResource (http://bioresource.nihr.ac.uk) and the NIHR Cambridge Biomedical Research Centre (BRC-1215-20014). The academic coordinating centre for INTERVAL was supported by core funding from the: NIHR Blood and Transplant Research Unit (BTRU) in Donor Health and Genomics (NIHR BTRU-2014-10024), NIHR BTRU in Donor Health and Behaviour (NIHR203337), UK Medical Research Council (MR/L003120/1), British Heart Foundation (SP/09/002; RG/13/13/30194; RG/18/13/33946) and NIHR Cambridge BRC (BRC-1215-20014; NIHR203312) [*]. Metabolon metabolomics assays in INTERVAL were funded by the: NIHR BioResource, NIHR Cambridge Biomedical Research Centre (BRC-1215-20014) [*], Wellcome Trust grant

number 206194 and BioMarin Pharmaceutical, Inc. The academic coordinating centre thank blood donor centre staff and blood donors for participating in the INTERVAL trial. This work was supported by Health Data Research UK, which is funded by the UK Medical Research Council, Engineering and Physical Sciences Research Council, Economic and Social Research Council, Department of Health and Social Care (England), Chief Scientist Office of the Scottish Government Health and Social Care Directorates, Health and Social Care Research and Development Division (Welsh Government), Public Health Agency (Northern Ireland), British Heart Foundation and Wellcome. For the purpose of open access, the author(s) has applied a Creative Commons Attribution (CC BY) licence to any Author Accepted Manuscript version arising from this submission.*The views expressed are those of the author(s) and not necessarily those of the NIHR, NHSBT or the Department of Health and Social Care. This work as supported by the National Health and Medical Research Council grants 1188493 and 1121474 (M.J.C.) Access to the TwinsUK cohort for this study was funded by a Hunter Medical Research Institute (Newcastle, Australia) Precision Medicine Research Program Pilot Grant (W.R.R). C.E.C. is supported by an investigator grant from the National Health and Medical Research Council (NHMRC). M.A.D. is supported by an investigator grant from the National Health and Medical Research Council (NHMRC). P.S. was supported by a Rutherford Fund Fellowship from the Medical Research Council (grant no. MR/S003746/1). We also acknowledge and thank all participants in the genetic studies included in this paper.

## Author contributions

W.R.R. designed the study and was the primary analyst. D.J.K. assisted with developing the phenome-wide Mendelian randomisation pipeline. M.A.D. developed the normative modeling pipeline. Z.F.G. performed the signature mapping analyses. P.S. performed quality control on the INTERVAL data prior to GWAS. K.K. performed the INTERVAL GWAS. E.D.C. and C.E.C. provided input into the TwinsUK retinol quality control, and the interpretation of the clinical associations of retinol. L.A.G. assisted with data curation and preparation of the manuscript. A.M.M. and D.A. provided the ATBC/PLCO data. M.J.C contributed to study design, funding, and manuscript preparation. All authors contributed to the interpretation of the results and the final manuscript.

## Competing interests

P.S. is now a full-time employee of GlaxoSmithKline. The remaining authors declare no competing interests.

## Additional information

[1]School of Biomedical Sciences and Pharmacy, The University of Newcastle, Callaghan, NSW, Australia. [2]Precision Medicine Research Program, Hunter Medical Research Institute, New Lambton, NSW, Australia. [3]Melbourne Neuropsychiatry Centre, Department of Psychiatry, The University of Melbourne, Melbourne, VIC, Australia. [4]Department of Anatomy and Physiology, The University of Melbourne, Melbourne, VIC, Australia. [5]Department of Psychiatry, Brigham and Women's Hospital, Harvard Medical School, Boston, MA, USA. [6]QIMR Berghofer Medical Research Institute, Brisbane, Queensland, Australia. [7]Human Genetics, Wellcome Sanger Institute, Wellcome Genome Campus, Hinxton, Cambridge, UK. [8]Department of Haematology, University of Cambridge, Cambridge Biomedical Campus, Cambridge, UK. [9]British Heart Foundation Cardiovascular Epidemiology Unit, Department of Public Health and Primary Care, University of Cambridge, Cambridge, UK. [10]British Heart Foundation Centre of Research Excellence, School of Clinical Medicine, Addenbrooke's Hospital, University of Cambridge, Cambridge, UK. [11]Health Data Research UK, Wellcome Genome Campus and University of Cambridge, Hinxton, UK. [12]School of Health Sciences, The University of Newcastle, Callaghan, NSW, Australia. [13]Food and Nutrition Research Program, Hunter Medical Research Institute, New Lambton, NSW, Australia. [14]Department of Epidemiology, University of Michigan School of Public Health, Ann Arbor, MI, USA. [15]Division of Cancer Epidemiology and Genetics, National Cancer Institute, NIH, Department of Health and Human Services, Bethesda, MD, USA. ✉e-mail: william.reay@uon.edu.au; murray.cairns@newcastle.edu.au

