## [Peer Review File · Nature Communications]

Genetic influences on circulating retinol and its relationship to human healthREVIEWER COMMENTS

Reviewer #1 (Remarks to the Author):

This is a nice and comprehensive study on retinol GWAS with a number of post GWAS analyses including extensive MR analysis.

- I was not confused on the GWAS study design re replication. You have a GWAS meta-analysis of 22K individuals and you choose to replicate on 5066 individuals? This is not very efficient. First of all, you replicated two previously reported loci. There is little point in replication those further. regarding the six novel loci, you could look at a combined meta-analysis across the three cohorts and add as additional criterion that there should be consistent direction of effect across all cohorts and at least $P < 0.01$ in each cohort and 5×10^{-8} in the combined meta-analysis. At the moment I understand that you looked at direction of effect only in the replication.

- Can you clearly highlight the novel loci in the tables. I wasn't sure what the bold rows of SNPs were. Which are the loci that were previously known, replicated and novel? Which ones do you find from one meta-analysis, the second or both? It is quite confusing now and the criteria are not well explained. Perhaps add a study design figure

- Throughout the mendelian randomisation analyses the reporting needs to improve to always mention GENETICALLY PREDICTED retinol is POTENTIALLY causal to phenotype X. At the moment the wording that retinol is causal is misleading.

- The MR sensitivity analyses may need more careful look. There may be a significant weighted median effect but if the MR egger estimate is largely affected this needs discussion as it may be pleiotropy

- The statistical significance threshold of the MR analysis is unclear, FDR 1% is stated in results but in methods it mentions nominal significance

- The phew on the finger outcome is much more interpretable and valid. Why not replicating in UK Biobank phewas as well? the 17000 outcomes is driven by continuous traits and GWAS of various quality and limitations.

- As the authors present MR with retinol as exposure and retinol as outcome it would make more sense to present this as bidirectional MR for all outcomes and discuss this based on the effects of bidirectional associations.

Reviewer #2 (Remarks to the Author):

This is an interesting further investigation into the biomolecular mechanisms driving retinol levels and its clinical importance on health outcomes. In general the paper is well written, although rather long in my humble opinion (but I will leave this to an editor to decide on). I feel that a number of the aspects (see below) are better placed in an online supplement. Please take notice on the following concerns and comments:

- I was a little surprised about the used study design, which did not have a more classical discovery-replication design, although the data was available for this. Now the authors make a somewhat vague statements on decrease in p-values etc. I would recommend to leave out the big meta-analysis of the 4 cohorts, and use the 2 cohorts added here as replication cohorts.

- My biggest concern with the presented results is the large heterogeneity between the research findings. Even though the meta-analysis in table 2 was done in 2 cohorts, 3 out of the 8 SNPs had a p-value for heterogeneity below 0.05 and 7 out of the 8 below 0.2 (suggesting at least there is some suggestive heterogeneity and evidence for heterogeneity in the biggest part of the research findings).

- The authors performed their Mendelian Randomization analyses on about 19,000 published studies that have publically-available GWAS data in the IEUGWAS database. There were no restrictions in size of ancestry (as far I can read). Please narrow down the list and only use the largest study available for the European ancestry; this will make more sense and makes sure current findings done now in smaller studies are not observed anymore in larger studies (and thus likely false-positive).

- Although already mentioned by the authors, I would be very cautious with the multi-instrument Mendelian Randomization analysis in this study; the vast majority of the instruments are well-known genes for other traits. GCKR is a highly pleiotropic gene. Same for FOXP2 (others I don't know by hard). This will certainly bias the results. Although perhaps with less statistical power, the Mendelian Randomization analyses will certainly be less biased with only the use of RBP4 as genetic instrument.

- The authors now use different sensitivity analyses for the multi-instrument Mendelian Randomization and make statements based on their p-values for statistical significance. Because MR-Egger is very much underpowered (much larger confidence intervals), this is not really an appropriate method; in case of MR-Egger, directional consistency works much better. Also, I would be careful with the weighted mode method, given the low number of instruments in this analysis.

Other comments:

- Please also add standard errors and mapped gene names to table 1 and 2.

- Please add a data on when the IEUGWAS database was used to derive all studies as this database is continuously updated

- There have been a number of Mendelian Randomization studies using retinol before, which haven't been mentioned in the paper. Please check.

Reviewer #3 (Remarks to the Author):

1. I would like to see more details about the design of the INTERVAL and METSIM cohorts including more details of the sample selection, demographic characteristics and how the assays were performed in order to assess the comparability.

2. The authors mention that SNP heritability was computed using the LDK-thin and BLD-LDK models. Could they provide more details on the methodology of how this was done?

3. In the gene prioritisation section the authors list the criteria for how genes were prioritised. This is mostly clear but the DAP-G method; CADD score and lowest RegulomeDB score require defining and further clarification for the unfamiliar reader.
4. It was not clear to me why the ARIC study was being used for the protein GRx analyses. The ARIC population is quite different from INTERVAL and METSIM populations used in terms of ancestry– so why is it being used here for these analyses? This requires clarification. I also did not see ARIC described in the data sources as well as FinnGen 8.
5. There are no details of the proteomic platform used to measure circulating proteins in plasma – this should be included. How many proteins were measured? Also not clear if the pQTLs identified from PWAS were cis or trans. If cis how defined? In terms of distance to the gene?
6. The authors mention that they conducted MR analyses for proteins using the IVW method. They do not describe any sensitivity analyses to assess the potential impact of horizontal pleiotropy – this should be included (as has been done for the MR-PheWAS).
7. As this is a complex design with many lines of investigation the manuscript would benefit from an overall diagram that illustrates the analysis pipeline and the key data sources that contributed to this pipeline.
8. More details on the information contained in the IEUGWASdb v6.9.2 would be helpful.
9. In the MR-PheWAS section the authors state that “We then repeated the above process of binary outcomes with at least 1138 1000 cases from FinnGen release 8...”. What was the minimum effect size that could be detected and what was the level of statistical power?
10. Can the authors speculate on why there were “markedly fewer variants” for the METSIM + INTERVAL+ATBC+PLCO meta-analysis than the METSIM+ INTERVAL meta-analysis? Unexplained heterogeneity?
11. The tables and figures could be presented more clearly and neatly. For example the number of events in Figure 3b should be reported.
12. There were several grammatical and typographical errors throughout the manuscript.

Response to reviewer comments

We thank the reviewers for their comments and feedback. We have responded to each reviewer comment in a point-by-point response detailed below, with changes to the revised manuscript highlighted in red.

REVIEWER #1

This is a nice and comprehensive study on retinol GWAS with a number of post GWAS analyses including extensive MR analysis.

- I was not confused on the GWAS study design re replication. You have a GWAS meta-analysis of 22K individuals and you choose to replicate on 5066 individuals? This is not very efficient. First of all, you replicated two previously reported loci. There is little point in replication those further. regarding the six new loci, you could look at a combined meta-analysis across the three cohorts and add as additional criterion that there should be consistent direction of effect across all cohorts and at least $P < 0.01$ in each cohort and 5×10^{-8} in the combined meta-analysis. At the moment I understand that you looked at direction of effect only in the replication.

We wish to clarify that our meta-analysis did in total include three datasets (METSIM, INTERVAL, and ATBC+PLCO), with our replication cohort of TwinsUK somewhat smaller than the number quoted by the reviewer. It is important to note that while approximately 5000 individuals were genotyped in TwinsUK, only ~1600 TwinsUK individuals had retinol measurements available at a minimum of one of the three measured timepoints (as now quoted in revised Figure 1). We then split these individuals into two unrelated cohorts (twin pairs separated) to maximise power at this sample size, resulting in up to 918 and 717 genotyped participants with measured retinol at three timepoints in each subset, respectively. As a result, we believe that it was appropriate to hold-out TwinsUK as the replication cohort as it was smaller than METSIM, INTERVAL or ATBC+PLCO. Our primary meta-analysis was METSIM+INTERVAL (N = 17,268, > 13 million variants), as well a larger sample-size meta-analysis with ATBC+PLCO that had less SNP coverage (< 4 million variants, N = 22,274). We chose to leave out TwinsUK so it could be used both as a replication for the lead SNPs, of which six of the eight were novel in this study, as well as for the polygenic scoring analyses. In the revised manuscript, we have included an overall schematic figure of the study design as figure 1 to enhance clarity for the reader. We chose to only report direction of effect in TwinsUK

given that each unrelated twin subset had a sample size less than 1000. However, one advantage of the TwinsUK cohort was that retinol was measured at three timepoints, allowing us to consider the mean effect size, and its direction, across these three timepoints. To clarify this further, we have included some additional text in the revised manuscript that denotes which of the lead SNPs were at least nominally significant in one or more of the measurements timepoints in TwinsUK (line 196).

- Can you clerkly highlight the novel loci in the tables. I wasn't sure what the bold rows of SNPs were. Which are the loci that were previously known, replicated and novel? Which ones do you find from one meta-analysis, the second or both? It is quite confusing now and the criteria are not well explained. Perhaps add a study design figure

We have edited table 1 and 2 to make it clearer to the reader that the bolded rows of the tables were the known retinol loci from the Mondul *et al.* GWAS for which evidence was further strengthened in this study. This was achieved by adding a new column that explicitly denoted which loci were novel relative to the previous dedicated retinol GWAS from Mondul *et al.*

We also agree with the reviewer that the overall clarity of the findings would be aided by adding a study design figure. We have constructed this as the new figure as *Figure 1* in the revised main text, which we hope will clarify some of these questions for the reviewer.

- Throughout the mendelian randomisation analyses the reporting need to improve to always mention GENETICALLY PREDICTED retinol is POTENTIALLY causal to phenotype X. At the moment the wording that retinol is causal is misleading.

We have further moderated the language used to describe the results of the Mendelian randomisation analyses in line with the reviewer's comment – for example, line 445, line 458, and line 525.

- The MR sensitivity analyses may need more careful look. There may be a significant weighted median effect but if the MR egger estimate is largely affected this needs discussion as it may be pleiotropy

We agree with the reviewer that the sensitivity analyses for the MR are particularly important for the interpretation of the 'polygenic' MR using all genome-wide significant signals as

instrumental variables (IVs). In our study, we used a tiering system whereby retinol – outcome trait pairings that survived multiple-testing correction ($FDR < 0.01$) using the screening model (inverse variance weighted estimator with multiplicative random effects) were subjected to a series of sensitivity analyses. These trait-pairings were then only retained which satisfied the following: no significant heterogeneity (Cochran’s Q , $P > 0.05$) in the exposure-outcome effect of each IV, leaving any single IV out did not ablate statistical significance, and the Egger intercept was not statistically significant. Thereafter, these traits were assigned a tier based the statistical significance of the remaining four MR methods. In the manuscript, we only reported retinol – outcome estimates with a minimum of two additional MR tests (besides the screening model) that were at least nominally significant ($P < 0.05$). As a result, whilst we agree with the reviewer that the MR-Egger method is an important component of our pipeline, we believe that our approach addresses these concerns since it does not rely on a particular test being significant but rather the existence multiple significant tests coupled with lack of evidence for heterogeneity and confounding pleiotropy. Another issue is that the MR-Egger approach is that it is the least powered of the five different MR methods we implemented, particularly with a relatively small number of IVs. This further necessitates an integrative approach which does not require statistical significance in any particular model in order to maintain sufficient power. We have included some additional text in the revised manuscript which outlines that reported trait pairings should be considered by the reader in light of assumptions made by individual models to further contextualise the rigour of these estimates (line 531).

- The statistical significance threshold of the MR analysis is unclear, FDR 1% is stated in results but in methods it mentions nominal significance

To clarify, we applied a statistical significance threshold for the primary MR model (IVW-MRE) was $FDR < 1\%$. Subsequently, for the sensitivity analyses, we then looked at which other MR models were nominally significant for the trait pairings that survived the above FDR threshold to assign a tier. This is also denoted in Figure 4A in the revised manuscript. We have additionally edited the wording of that section to enhance clarity for the reader.

- The phew on the finger outcome sis much more interpretable and valid. Why not replicating in UK Biobank phewas as well? the 17000 outcomes is driven by continuous traits anode GWAS of various quality and limitations.

We wish to clarify that thousands of UK Biobank GWAS, both from the Neale group and the IEUGWAS run UK Biobank GWAS, were included in our IEUGWAS based pheWAS that we report in the manuscript. We have edited the revised text to make this clearer for the reader (line 456). We chose FinnGen release 8 as our second primary data source for the pheWAS as these GWAS were performed using a Firth regression-based mixed model approach that increases power in the situation of case/control imbalance, which is a common problem when analysing binary phenotypes in population biobanks.

- A the authors present MR with retinol as exposure and retinol as puctoem it would make more sense to present this as bidirectional MR for all outcomes and discussthsi based on the effects of bidirectional associations.

Whilst we agree that the consolidating the MR findings in this way can be useful, the complexity of our MR pipeline would render a single section describing all findings together overly long and complex. We explicitly consider the existence of any bidirectional effects in the manuscript for any traits with strong evidence for a forward effect of retinol (line 715) or a reverse effect on retinol (line 720), respectively. We decided the best way was to proceed was to include an additional subheading which combines any discussion of any potential bidirectional findings in an integrated fashion (line 712). We decided on this approach given there was limited to no evidence of any traits which exhibit a bidirectional relationship with circulating retinol as uncovered by our pipeline.

REVIEWER #2

This is an interesting further investigation into the biomolecular mechanisms driving retinol levels and its clinical importance on health outcomes. In general the paper is well written, although rather long in my humble opinion (but I will leave this to an editor to decide on). I feel that a number of the aspects (see below) are better placed in an online supplement. Please take notice on the following concerns and comments:

- I was a little suprised about the used study design, which did not have a more classical discovery-replication design, although the data was available for this. Now the authors make

a somewhat vague statements on decrease in p-values etc. I would recommend to leave out the big meta-analysis of the 4 cohorts, and use the 2 cohorts added here as replication cohorts.

We acknowledge the reviewer's feedback and have made considerable efforts to enhance the clarity of the study's design in our revised manuscript. We wish to clarify that we did implement a discovery-replication design in this study – however, we have endeavoured in the revised version of the manuscript to make the overall study design and workflow clearer to the reader. One of the main activities related to this was the inclusion of a new overall schematic figure of the study design (revised figure 1).

We performed two primary meta-analyses. Firstly, we meta-analysed the METSIM sample with INTERVAL (> 13 million total variants, N = 17,268), followed by a larger sample-size meta-analysis with ATBC+PLCO added that had less SNP coverage (< 4 million total variants, N = 22,274). These two strategies were adopted to balance excellent genome-wide variant coverage (METSIM+INTERVAL) with maximal sample size (METSIM+INTERVAL+ATBC+PLCO). We explicitly tested for heterogeneity in each of these meta-analyses, as outlined in response to the reviewer's next comment. In the manuscript, we also directly reported how the association of the lead SNPs changed upon the addition of ATBC+PLCO, as well as now visualising the association within each contributing study individually in the new Supplementary Figure 6, which has been added to the supplementary material of the revised manuscript. The preceding text all related to our discovery analysis, with the TwinsUK cohort then used as a formal replication sample. As outlined in text, TwinsUK had a somewhat smaller sample size than any of the other cohorts with measured retinol available. Therefore, we reported both the lead SNPs that were at least nominally significant in TwinsUK (line 196) and whether the number of lead SNPs with a consistent direction of effect on retinol in TwinsUK relative to the discovery meta-analyses was greater than chance alone using a Binomial test (line 193).

- My biggest concern with the presented results is the large heterogeneity between the research findings. Even though the meta-analysis in table 2 was done in 2 cohorts, 3 out of the 8 SNPs had a p-value for heterogeneity below 0.05 and 7 out of the 8 below 0.2 (suggesting at least there is some suggestive heterogeneity and evidence for heterogeneity in the biggest part of the research findings).

We agree with the reviewer that a more comprehensive investigation of any heterogeneity between the contributing studies is warranted. In response, we have conducted a thorough exploration of the heterogeneity within these studies, which has been incorporated into the revised manuscript. Specifically, we have included some additional supplementary figures which demonstrate that heterogeneity is not a particularly problematic issue in terms of the signals that we identify in this study (Supplementary Figure 3,4, and 6). As outlined in the revised supplementary figure 3 (see below), the per SNP effects of each lead signal in METSIM and INTERVAL are quite similar in the sense that their confidence intervals all overlap. A similar observation is also evident considering the Z-score in each cohort.

An analogous plot for supplementary figure 3 was also included for the METSIM+INTERVAL+ATBC+PLCO meta-analysis. One additional thing to note is that as the primary meta-analysis was sample size weighted, the more precise SNP-estimates from INTERVAL were given the most weight in the meta-analysis.

Supplementary Figure 3. Per-cohort effects of lead SNPs (METSIM+INTERVAL meta-analysis). On the left-hand side, the effect size in SD (95% confidence intervals) for the lead SNPs in each of the cohorts. On the right-hand side, the retinol Z score (beta divided by standard error) in each cohort.

- The authors performed their Mendelian Randomization analyses on about 19,000 published studies that have publically-available GWAS data in the IEUGWAS database. There were no restrictions in size of ancestry (as far I can read). Please narrow down the list and only use the largest study available for the European ancestry; this will make more sense and makes sure current findings done now in smaller studies are not observed anymore in larger studies (and thus likely false-positive).

We understand the reviewers concerns and agree that this is a limitation of using a phenome-wide approach that leverages a large resource such as IEUGWASdb. The task of identifying the ‘optimal’, that is newest and largest GWAS, for thousands of phenotypes *a priori* is very difficult due to inconsistent phenotype naming, and at times, incomplete information from IEUGWAS. As a result, we believe that these issues in the context of this study are best addressed during the ‘tiering’ process wherein we prioritise the trait pairings that display the most confident evidence of a causal effect of circulating retinol through our suite of integrative sensitivity analyses. In light of the reviewer’s concerns, we have manually reviewed all the tier #2 and tier #3 and report if larger GWAS exist, as well as their ancestry was not primarily European (line 515). We also then updated figure 4 in the revised text to only report tier #2/#3 traits which represent large sample size (for that trait), European ancestry GWAS reported in the IEUGWAS database. We also decided to remove any FinnGen release 5 traits from the tiering output. FinnGen release 5 does contain some phenotype definitions not used in release 8; however, as release 8 was specifically analysed in the subsequent section, we believe it is more appropriate to not report these release 5 results as tier #2 or tier #3 traits. In terms of the microbiome outcome phenotypes, a larger, more recent GWAS does exist in the literature than what is available in IEUGWAS, however, it overlaps in sample with our GWAS, and therefore, we chose not to analyse these data. A future direction would be to replicate our findings in newer GWAS when they emerge, which we now explicitly state in text in the revised version (line 526).

- Although already mentioned by the authors, I would be very cautious with the multi-instrument Mendelian Randomization analysis in this study; the vast majority of the instruments are well-known genes for other traits. GCKR is a highly pleiotropic gene. Same for FOXP2 (others I don't know by hard). This will certainly bias the results. Although perhaps with less statistical power, the Mendelian Randomization analyses will certainly be less biased with only the use of RBP4 as genetic instrument.

We agree with the reviewer that this is an important consideration and, as a result, have restructured the way our Mendelian randomisation results are reported in the revised manuscript. Specifically, we now begin the Mendelian randomisation text in the results with a separate section that more explicitly describes the *RBP4* estimates as the most confident results (line 438), along with a dedicated figure in text (Figure 3 in the revised text). The *RBP4* IV is also now applied in the revised version to FinnGen release 8. We then report the ‘polygenic’ MR in the next subheading, further emphasising that due to highly pleiotropic IVs, like that in *GCKR*, these results must be interpreted very carefully (line 494). The correlation between the single *RBP4* IV and the IVW (all IVs) results for FinnGen r8 ($r = 0.48$) is also now reported in that section. The FinnGen figure from the original manuscript has now been moved to the supplementary so there is no undue emphasis placed on the MR analyses using all IVs, resulting in a balance in the number of in text figures (one *RBP4* IV figure – revised figure 3 and one figure for the results using all IVs – revised figure 4). Crucially, we also specifically state in the text the concordance with the *RBP4* IV derived direction and magnitude for the most confident results using all IVs.

- The authors now use different sensitivity analyses for the multi-instrument Mendelian Randomization and make statements based on their p-values for statistical significance. Because MR-Egger is very much underpowered (much larger confidence intervals), this is not really an appropriate method; in case of MR-Egger, directional consistency works much better. Also, I would be careful with the weighted mode method, given the low number of instruments in this analysis.

We agree with the reviewer that the low power of the MR-Egger approach is an important consideration. However, we chose the more conservative approach of testing the significance of the MR-Egger model given that, as the reviewer mentions above, some of the retinol IVs like that variant in *GCKR* are highly pleiotropic. In our integrated approach, we decided to not predicate our tiering system used for assigning strength of evidence on a single MR test, but rather consistency across multiple different methods and lack of strong evidence for heterogeneity/confounding pleiotropy. We have explicitly highlighted in the revised manuscript that one limitation of our pipeline is the expected underpowered nature of the MR-Egger method (line 531).

Other comments:

- Please also add standard errors and mapped gene names to table 1 and 2.

To enhance clarity of the table for the reader, table 1 only now reports the Stouffer Z score given that is the GWAS used to define the lead SNP, the beta and standard error for lead SNPs can be found in the supplementary summary material. The closest gene (transcription start site) has now been added to the revised table 1 and 2.

- Please add a data on when the IEUGWAS database was used to derive all studies as this database is continuously updated

We have added this information to the methods of the revised manuscript on line 1148.

- There have been a number of Mendelian Randomization studies using retinol before, which haven't been mentioned in the paper. Please check.

We agree with the reviewer that further discussion of previous Mendelian randomisation studies using retinol is warranted. In the revised manuscript, we have included some additional discussion of these studies starting on line 854.

REVIEWER #3

1. I would like to see more details about the design of the INTERVAL and METSIM cohorts including more details of the sample selection, demographic characteristics and how the assays were performed in order to assess the comparability.

We have added some additional text to the methods which further compares these two cohorts (e.g. line 888, 901). Moreover, a more extensive explanation of the INTERVAL cohort has been included in the supplementary text on page 27. As now more explicitly stated in the revised text, both cohorts consist of recruited volunteers for which plasma retinol was quantified using the same instrument [DiscoveryHD4[®] (Metabolon, Inc., Durham, USA)]. As this was a meta-analysis, we were able to explicitly test for any heterogeneity which may unduly influence the genetic findings. In response to a comment from reviewer #2, we have added some additional

text on exploring heterogeneity between these cohorts, as well some additional supplementary figures (Supplementary Figures 3,4, and 6).

2. The authors mention that SNP heritability was computed using the LDAK-thin and BLD-LDAK models. Could they provide more details on the methodology of how this was done?

We have now included some additional text further describing these two heritability models in the SNP heritability section of the methods (line 1021).

3. In the gene prioritisation section the authors list the criteria for how genes were prioritised. This is mostly clear but the DAP-G method; CADD score and lowest RegulomeDB score require defining and further clarification for the unfamiliar reader.

In the revised text, we have elaborated on the description of these three components of the prioritisation pipeline to enhance clarity for the reader (line 1054-1061). We specifically explained the rationale for using the DAP-G method, as well as what both the CADD score and RegulomeDB score represent.

4. It was not clear to me why the ARIC study was being used for the protein GReX analyses. The ARIC population is quite different from INTERVAL and METSIM populations used in terms of ancestry— so why is it being used here for these analyses? This requires clarification. I also did not see ARIC described in the data sources as well as FinnGen 8.

We wish to clarify that pQTLs/PWAS models were only derived from the European subset of ARIC, as described by Zhang, Chatterjee *et al.*, 2022 Nature Genetics. We have edited the methods to make this clear to the reader (line 1109, 1116). ARIC and FinnGen release 8 have also now been added to the data sources in the revised manuscript.

5. There are no details of the proteomic platform used to measure circulating proteins in plasma – this should be included. How many proteins were measured? Also not clear if the pQTLs identified from PWAS were cis or trans. If cis how defined? In terms of distance to the gene?

We have included these additional details in the methods section of the revised manuscript (line 1117). Briefly, the proteomic platform used was the SomaLogic Inc. version 4 platform via an aptamer (SOMAmer) based approach. In the ARIC study, 4,697 unique proteins or protein complexes were measured. In our study, we focused on finemapped pQTLs (posterior probability > 0.5) within 500 kb of the transcription start site. After harmonisation with the retinol GWAS, 937 unique proteins or protein complexes were able to be tested.

6. The authors mention that they conducted MR analyses for proteins using the IVW method. They do not describe any sensitivity analyses to assess the potential impact of horizontal pleiotropy – this should be included (as has been done for the MR-PheWAS).

For the proteomic MR analyses, practically all proteins only had a single IV available. This was due to only using finemapped pQTLs as IVs that had evidence for a causal relationship with protein abundance via finemapping, as described in response to a comment above. As a result, we are not able to perform the same suite of analyses as in the MR-pheWAS that used all retinol associated SNP as IVs. Instead, we used colocalisation to test the posterior probability of a single causal variant driving the association with both the protein and retinol. If the same causal variant is linked to both, this greatly reduces the possibility of confounding due to factors like horizontal pleiotropy. However, as using a single IV is limited in terms of available sensitivity analyses, we have added a note of caution in that section of the revised manuscript (line 419).

7. As this is a complex design with many lines of investigation the manuscript would benefit from an overall diagram that illustrates the analysis pipeline and the key data sources that contributed to this pipeline.

We agree with the reviewer and have constructed an overall schematic figure in the revised manuscript (revised figure 1).

8. More details on the information contained in the IEUGWASdb v6.9.2 would be helpful.

We have added some additional information regarding IEUGWASdb in the revised manuscript, further describing that it reports SNP effect sizes and statistical significance for thousands of phenotypes subjected to GWAS (line 456, line 1149).

9. In the MR-Phewas section the authors state that “We then repeated the above process of binary outcomes with at least 1138 1000 cases from FinnGen release 8...”. What was the minimum effect size that could be detected and what was the level of statistical power?

A few different approaches for power calculation approaches with respect to MR have been previously reported in the literature. We did not consider a power analysis in our approach because of the nature of our analyses being phenome-wide, that is, testing the effect of retinol on over 1000 binary outcomes without a prior hypothesis. It would have been impractical to tune the parameters of a power analysis for this many phenotypes in a way that is meaningful. We also believe that a post-hoc power analysis would not be appropriate given our rationale for not including this *a priori*.

10. Can the authors speculate on why there were “markedly fewer variants” for the METSIM + INTERVAL+ATBC+PLCO meta-analysis than the METSIM+ INTERVAL meta-analysis? Unexplained heterogeneity?

The discrepancy in the number of available variants for the ATBC+PLCO cohort, as opposed to INTERVAL and METSIM, can be attributed to the genotyping platform and imputation panel used. Specifically, ATBC/PLCO samples were genotyped using the Illumina HumanHap550/610 arrays and imputed to the HapMap Central European reference panel. A limitation of this is that 600,000 variants were available for inclusion in the GWAS, as was common at the time (2010) before more recent advances in imputation pipelines that result in larger post-imputation yield. Therefore, to increase the number of variants available for meta-analysis, we applied a summary statistics-based imputation procedure to boost the number of variants available for meta-analysis. Even after summary statistics imputation boosted the number of variants to ~4 million – this was significantly less than METSIM and INTERVAL. INTERVAL used whole genome sequencing, whilst METSIM was able to leverage a Finnish imputation backbone that performs very well in that population in terms of the number of confidently imputed variants. In summary, the reason why the METSIM+INTERVAL+ATBC+PLCO meta-analysis had markedly fewer variants is due to the technical limitations at the time the ATBC+PLCO GWAS was performed (2010). This was the why we chose to perform both the METSIM+INTERVAL meta-analysis such that genome-wide variant coverage was optimal, as well as METSIM + INTERVAL+ATBC+PLCO, which has a larger sample size.

11. The tables and figures could be presented more clearly and neatly. For example the number of events in Figure 3b should be reported.

We have endeavoured to improve presentation and clarity throughout the revised manuscript. For example, we have added more information to table 1 and 2 related to the closest gene and specific demarcation whether the signal is novel, as requested by the other two reviewers. We also corrected the formatting issue with the Manhattan plots in revised Figure 2a and 2b, so they are aligned by only plotting the autosomal signals. Figure 3b in the original text is now supplementary figure 9 in the revised manuscript, however, we have added the number of events for each phenotype in the revised version of that figure. That figure was moved to the supplement in response to the fourth comment of reviewer 2. We have also made the colour scheme of revised figure 4 different between the two sets of tiers included to enhance clarity for the reader in terms of the different tiers, as well improving overall presentation.

12. There were several grammatical and typographical errors throughout the manuscript.

We thank the reviewer for alerting us to these. In the revised manuscript, we have endeavoured to correct these throughout the text where they were found.

REVIEWERS' COMMENTS

Reviewer #1 (Remarks to the Author):

The authors have addressed sufficiently the comments raised by the 3 reviewers.

Reviewer #2 (Remarks to the Author):

The authors addressed my concerns

Reviewer #3 (Remarks to the Author):

I am happy with the responses to my previous comments and the revisions made to the manuscript.